# Mode-1 $N_2$ internal tides observed by satellite altimetry

Zhongxiang Zhao[1, 2]

[1]Applied Physics Laboratory, University of Washington, Seattle, Washington, USA
[2]School of Oceanography, University of Washington, Seattle, Washington, USA

**Correspondence:** Zhongxiang Zhao (zzhao@uw.edu)

**Abstract.** Satellite altimetry provides a unique technique for observing the sea surface height (SSH) signature of internal tides from space. Previous studies have constructed empirical internal tide models for the four largest constituents $M_2$, $S_2$, $K_1$ and $O_1$ by satellite altimetry. Yet no empirical models have been constructed for minor tidal constituents. In this study, we observe mode-1 $N_2$ internal tides (the fifth largest constituent) using about 100 satellite-years of SSH data from 1993 to 2019. We employ a recently-developed mapping procedure that includes two rounds of plane wave analysis and a two-dimensional bandpass filter in between. The results show that mode-1 $N_2$ internal tides have millimeter-scale SSH amplitudes. Model errors are estimated from background internal tides that are mapped using the same altimetry data, but with a tidal period of 12.6074 hours ($N_2$ minus 3 minutes). The global mean error variance is about 25% that of $N_2$, suggesting that the mode-1 $N_2$ internal tides can overcome model errors in some regions. We find that the $N_2$ and $M_2$ internal tides have similar spatial patterns, and that the $N_2$ amplitudes are about 20% of the $M_2$ amplitudes. Both features are determined by the $N_2$ and $M_2$ barotropic tides. The mode-1 $N_2$ internal tides are observed to propagate hundreds to thousands of kilometers in the open ocean. The globally integrated $N_2$ and $M_2$ internal tide energies are 1.8 and 30.9 PJ, respectively. Their ratio of 5.8% is larger than the theoretical value of 4%, because the $N_2$ internal tides contain relatively larger model errors. Our mode-1 $N_2$ internal tide model is evaluated using independent satellite altimetry data in 2020 and 2021. The results suggest that the model can make internal tide correction in regions where the model variance is greater than twice the error variance. This work demonstrates that minor internal tidal constituents can be observed using multiyear multisatellite altimetry data and dedicated mapping techniques.

## 1 Introduction

The Moon's elliptical orbit around the Earth has an eccentricity of $\approx$0.055, with its perigean and apogean distances being about $3.63\times10^5$ and $4.06\times10^5$ km, respectively. The Moon completes one revolution every 27.5546 days (one anomalistic month). The tidal constituents $L_2$ and $N_2$ are induced by the Moon's elliptical orbit (Doodson, 1921). They are named the smaller and larger lunar elliptical semidiurnal constituents. The $L_2$ and $N_2$ periods are 12.1916 and 12.6583 hours, respectively (Doodson, 1921; Pawlowicz et al., 2002). $M_2$ (12.4206 hours) is based on the mean distance between the Earth and the Moon ($3.84\times10^5$ km). While the $L_2$ and $N_2$ superposition gives the 27.5546-day perturbation with the Moon-Earth distance changes along the elliptic orbit. On global average, the amplitudes of $M_2$, $N_2$, and $L_2$ have a respective ratio of 1 : 0.2 : 0.05. $N_2$ is the fifth largest tidal constituent; therefore, its impact on the ocean environment is significant. For example, in waters around New Zealand, the $N_2$ barotropic tide has larger amplitudes than $S_2$ (Byun and Hart, 2020, Figure 4 therein). The superposition of $N_2$, $M_2$, $L_2$,

and $S_2$ can cause perigean spring tides (king tides) and apogean neap tides, which significantly affect harbors, coastal regions, and estuaries (Wood, 1978). Including $N_2$ internal tides can simulate the temporal variation of internal tide energetics with the Moon's elliptical motion. Theoretically, $N_2$ may modulate $M_2$ internal tides by $\pm20\%$ in amplitude, and by $\pm40\%$ in energy
(i.e., $(1\pm0.2)^2$). On average, $N_2$ will enhance the $M_2$-induced ocean mixing by 4% (i.e., $0.2^2$).

Internal tides are widespread in the ocean and affect numerous ocean processes such as diapycnal mixing, tracer transport and acoustic transmission (Wunsch, 1975; Dushaw et al., 1995; Whalen et al., 2020). Internal tides may provide about half of the power for diapycnal mixing in the ocean interior (Munk and Wunsch, 1998; Egbert and Ray, 2000; MacKinnon et al., 2017; Kelly et al., 2021). The magnitude and geography of diapycnal mixing may modulate the large-scale ocean circulation
and global climate change; therefore, it is important to study their generation, propagation, and dissipation in the global ocean (Jayne and St. Laurent, 2001; Melet et al., 2016; Pollmann et al., 2019; Vic et al., 2019; de Lavergne et al., 2020; Arbic, 2022). Internal tides are annoying noise in the study of mesoscale and sub-mesoscale dynamics. In particular, it will be necessary to make internal tide correction to the Surface Water and Ocean Topography (SWOT) data, so that one can better study the sub-mesoscale dynamic processes (Fu and Ubelmann, 2014; Qiu et al., 2018; Wang et al., 2018; Morrow et al., 2019). Empirical
internal tide models can be constructed using past satellite altimetry sea surface height (SSH) measurements. However, previous satellite observations focus mainly on the four largest tidal constituents $M_2$, $S_2$, $K_1$ and $O_1$ (Dushaw, 2015; Ray and Zaron, 2016; Zhao et al., 2016; Zaron, 2019; Ubelmann et al., 2022). Dushaw (2015) attempts to map $N_2$ internal tides using the TOPEX/Poseidon data from 1992 to 2008, but fails to obtain an empirical model, because the resulting $N_2$ internal tides are too noisy (see his Figures 38 and 52). That work is mainly limited by the short data set available then. In this study, we will
construct a reliable empirical $N_2$ internal tide model using a larger data set and a recently-developed mapping method.

The challenge of observing $N_2$ internal tides by satellite altimetry lies in their small SSH displacements (Dushaw, 2015). Given that $M_2$ internal tides have SSH amplitudes of 1–2 cm, $N_2$ internal tides have only sub-centimeter SSH amplitudes. In this paper, the observation of $N_2$ internal tides is made possible by two improvements. First, a larger SSH data set is available, thanks to almost three decades of multiple satellite observations since 1993. The merged data set from 1993 to 2019 is about 100
satellite-years long; therefore, nontidal errors can be significantly suppressed. Second, a recently-developed mapping procedure is employed. This mapping technique extracts $N_2$ internal tides utilizing their known frequency and theoretical wavelengths. Nontidal errors can be significantly suppressed by both temporal and spatial filters. The resulting $N_2$ internal tides reveal their basic features in the global ocean, although they are still noisy (compared to the much larger $M_2$ internal tides). It is challenging (though possible) to extract $L_2$ internal tides in some regions, which are estimated to have 1-mm SSH signals at most (5% of
$M_2$).

The rest of this paper is arranged as follows. Section 2 describes the data and methods used in this paper. Section 3 presents and discusses the new $N_2$ internal tides, mainly by comparing with the well-studied $M_2$ internal tides. Section 4 is a summary.

## 2 Data and methods

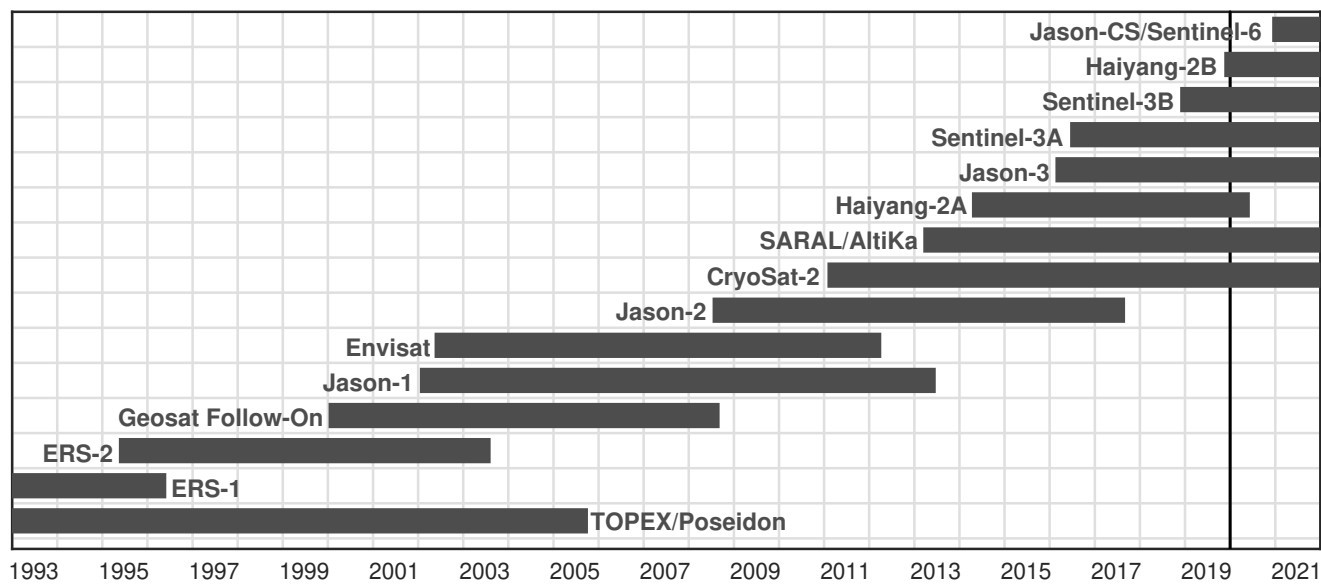

**Figure 1.** Satellite altimetry data used in this study. The merged multiyear multisatellite data from 1993 to 2019 are about 100 satellite-years long. The altimetry data in 2020 and 2021 are taken as independent data for model evaluation.

## 2.1 Data

The satellite altimetry SSH data used in this paper are collected by multiple altimetry missions from 1993 to 2021. In the order of launch time, they are TOPEX/Poseidon, ERS-1, ERS-2, Geosat Follow-On, Jason-1, Envisat, Jason-2, CryoSat-2, SARAL/AltiKa, Haiyang-2A, Jason-3, Sentinel-3A, Sentinel-3B, Haiyang-2B and Jason-CS/Sentinel-6 (Figure 1). The combined data set from 1993 to 2019 is about 100 satellite-years long. We use the satellite along-track SSH data downloaded from the Copernicus Marine Service (https://doi.org/10.48670/moi-00146). The SSH measurements have been processed by standard

corrections for atmospheric effects, surface wave bias, and geophysical effects (Pujol et al., 2016; Taburet et al., 2019). The ocean barotropic tide, polar tide, solid Earth tide and loading tide are corrected using theoretical or empirical models (Pujol et al., 2016). Mesoscale correction (Ray and Byrne, 2010; Zhao, 2022a) is made using the gridded SSH fields downloaded from the Copernicus Marine Service (https://doi.org/10.48670/moi-00148). The satellite along-track SSH data in 2020 and 2021 are used to evaluate the new $N_2$ internal tide model as independent data (Section 2.6). Extracted from the 27-year-long data, our

$N_2$ internal tide model contains only the 27-year-coherent component. Their temporal variation (or incoherent component) is not addressed in this paper.

The observation of internal tides by satellite altimetry may be affected by an issue called tidal aliasing, because the satellite repeat cycles are much longer than the semidiurnal and diurnal tidal periods. Here we examine possible tidal aliasing issues with $N_2$ and $M_2$ internal tides. In one 160 km by 160 km fitting window (section 2.2), there are typically about $7.84 \times 10^4$

SSH data from multiyear multisatellite measurements. Using their observation times, we can calculate their phase lags with respect to the $N_2$ tidal cycle (12.6583 hours). Figure 2a gives the histogram of their phase lags over one $N_2$ tidal cycle. For

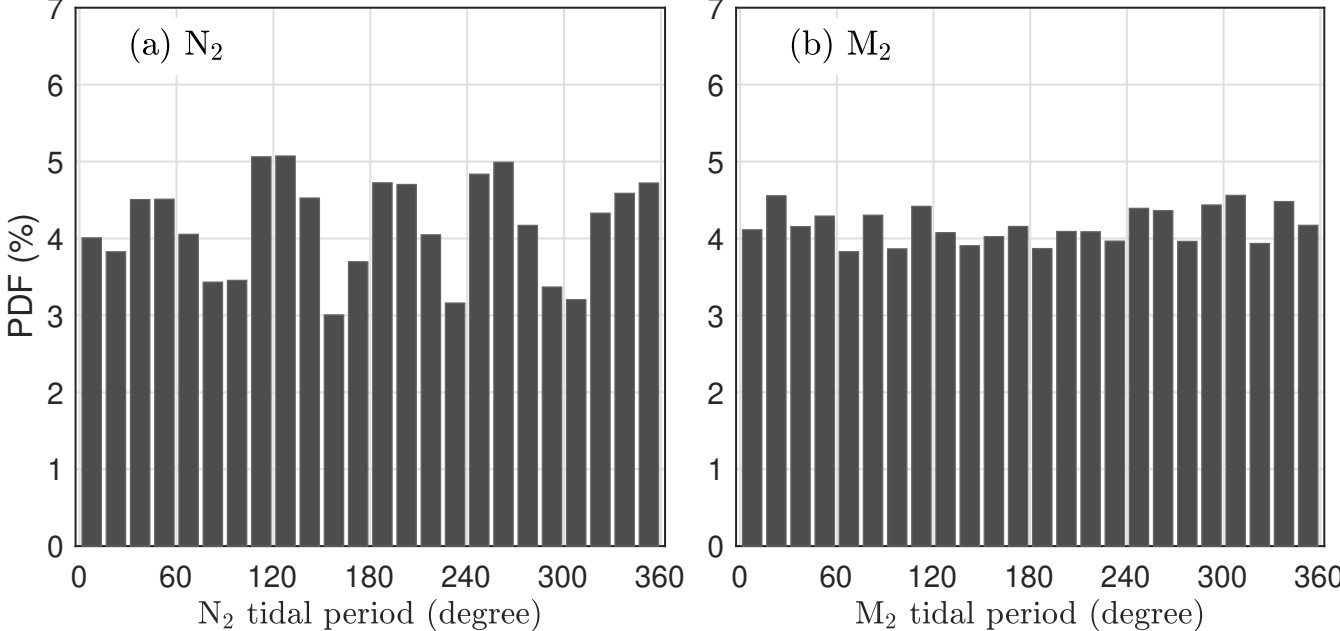

**Figure 2.** Histograms of the phase lags of the satellite SSH data over one $N_2$ (a) and $M_2$ (b) tidal period. This example is for one 160 km by 160 km window centered at 15°N, 160°W. There are about $7.84 \times 10^4$ SSH measurements in this window. From their observation times, the phase lags with respect to $N_2$ and $M_2$ are calculated.

comparison, Figure 2b shows the histogram with respect to the $M_2$ tidal cycle (12.4206 hours). The results show that the SSH data overall evenly distribute over one $N_2$ or $M_2$ tidal cycle, without extreme biases. In particular, their distribution on $M_2$ is smooth, suggesting that there is no tidal aliasing issues for $M_2$. Their distribution over $N_2$ is a little bumpy, suggesting that the resulting $N_2$ internal tides may have larger errors. The uneven distribution stems from the orbital configurations of the satellite missions. Fortunately, as shown in this study, the new mode-1 $N_2$ internal tides can overcome background noise in some regions.

### 2.2 Plane wave analysis

The core technique of our mapping procedure is plane wave analysis. By this method, internal tides are determined by fitting horizontal plane waves in one given fitting window (160 km by 160 km in this study), in contrast to harmonic analysis at one single site. This method has been described in detail in our previous studies (Zhao and Alford, 2009; Zhao, 2014; Zhao et al., 2016). For each tidal constituent, there may be multiple internal tides of arbitrary propagation directions at each site, due to their multiple source regions and long-range propagation. Plane wave analysis can resolve these internal tides by propagation direction. We will fit 5 mode-1 $N_2$ internal tidal waves at each site. Our 5-wave representor follows

$$\Sigma_{m=1}^{5} A_m \cos(k\,x\cos\theta_m + k\,y\sin\theta_m - \omega\,t - \phi_m), \tag{1}$$

where $x$ and $y$ are the east and north Cartesian coordinates, $t$ is time, $\omega$ and $k$ are the frequency and horizontal wavenumber of the target internal tides, respectively. Three parameters need to be determined for each internal tidal wave: amplitude $A$, phase $\phi$, and direction $\theta$. To determine one wave, the amplitude and phase of a single plane wave are determined by the least-squares fit in each compass direction (with $1°$ increment). When the resulting amplitudes are plotted as a function of direction in polar coordinates, an internal tidal wave appears to be a lobe. The direction of the first wave is thus determined from the biggest lobe. Thus, the amplitude $A$, phase $\phi$, and propagation direction $\theta$ of one internal tidal wave are determined. Afterward, its signal is predicted and subtracted from the original data, which removes the wave itself and its side lobes. This procedure can be repeated to extract an arbitrary number of waves one by one. The resulting internal tidal waves are sorted with descending amplitudes.

The frequency ($\omega$) and horizontal wavenumber ($k$) of the target internal tides are needed in plane wave analysis. The $M_2$ and $N_2$ tidal periods (equivalent to frequencies) are from the Moon's orbital motion around the Earth (Doodson, 1921; Pawlowicz et al., 2002). They are 12.4206 and 12.6583 hours, respectively. The local phase speed (equivalent to wavenumber) of the target internal tides is theoretically determined from the World Ocean Atlas 2018 (WOA18) (Boyer et al., 2018). The WOA18 provides climatological hydrographic profiles on a spatial grid of $0.25°$ latitude by $0.25°$ longitude. Ocean depth is based on the 1-arc-minute topography database constructed using in situ and satellite measurements (Smith and Sandwell, 1997). For given ocean depth and stratification profile, the vertical structures and eigenvalue speeds of internal tides are obtained by solving the Sturm-Liouville equation (Gill, 1982; Chelton et al., 1998; Kelly, 2016)

$$\frac{d^2\Phi(z)}{dz^2} + \frac{N^2(z)}{c^2}\Phi(z) = 0, \tag{2}$$

subject to free-surface and rigid-bottom boundary conditions, where $N(z)$ is buoyancy frequency profile, and $c$ is the eigenvalue speed. The phase speed $c_p$ can be calculated from the eigenvalue speed following

$$c_p = \frac{\omega}{\sqrt{\omega^2 - f^2}}c, \tag{3}$$

where $\omega$ and $f$ are the tidal and inertial frequencies, respectively. Note that the phase speed is a function of longitude and latitude (Zhao et al., 2016).

## 2.3 Mapping procedure

Our three-step mapping procedure consists of two rounds of plane wave analysis and a spatial two-dimensional (2D) bandpass filter in between (Zhao, 2020, 2021, 2022a, b). In this paper, the mapping process is illustrated by showing intermediate results in Figure 3. An interested reader is referred to the above papers for more details.

In step 1, mode-1 $N_2$ internal tides are mapped by plane wave analysis as described above. The $N_2$ internal tides are mapped from along-track SSH data onto a spatially regular grid. In this paper, our fitting window is chosen to be 160 km by 160 km, consistent with wavelengths of mode-1 $N_2$ internal tides. The resulting $N_2$ internal tides are at a $0.2°$ longitude by $0.2°$ latitude grid. At each grid point, five mode-1 $N_2$ internal tidal waves of arbitrary propagation directions are determined. The vector sum of these five waves gives the internal tide solution. Figure 3a shows the mode-1 $N_2$ internal tide field obtained in this step.

It gives obvious internal tide signals (e.g., around the Hawaiian Ridge), but the nontidal noise is high. In step 2, the spatially regular $N_2$ internal tide field is cleaned by a 2D bandpass filter in overlapping 850 km by 850 km windows. The $N_2$ internal tide field is first converted to the 2D wavenumber spectrum by Fourier transform. The spectrum is truncated to [0.8 1.25] times the local wavenumber. The truncated spectrum is converted back to the internal tide field by inverse Fourier transform. Figure 3b shows the cleaned $N_2$ internal tide field. Now the $N_2$ internal tide signals are much cleaner. However, Figure 3b cannot resolve multiple internal tidal waves yet. In step 3, plane wave analysis is called again to decompose the filtered internal tide field into five internal waves of arbitrary propagation directions. The second-round plane wave analysis is same as the first-round plane wave analysis, except that the input is the filtered internal tide field in step 2. In the end, the resulting five waves are saved with their respective amplitudes, phases and directions. Figure 3c shows the 5-wave superimposed internal tide field. It is very similar to Figure 3b, because this step only decompose the internal tide field. The 5-wave decomposition allows us to separate internal tides of different propagation directions. They will be used to extract long-range internal tidal beams in the ocean (Section 3).

## 2.4 $N_2$ and $M_2$ internal tides

We map both the mode-1 $N_2$ and $M_2$ internal tides following the same 3-step procedure. They are constructed from the same satellite altimetry data, but using their respective wave parameters (frequency and wavenumber). Figure 4 shows the resulting $N_2$ and $M_2$ internal tide fields. Internal tides in shallow waters ($<$1000 m) are discarded. The new $M_2$ internal tides are almost identical to those obtained in previous studies using slightly different satellite data (Zhao, 2022b). Here we find that the $N_2$ and $M_2$ internal tides have similar spatial patterns, and that the $N_2$ amplitudes are about 20% of the $M_2$ amplitudes. The largest $N_2$ amplitudes are about 5 mm, compared to 20–30 mm for $M_2$ internal tides. To account for this factor, their colormap ranges are different by a factor of 5. Figure 4 gives SWOT ground tracks in its one-day fast-repeating phase (green lines). It shows that strong mode-1 $N_2$ internal tides occur under some SWOT swaths, for example, those off the California coast, in the New Caledonia region, in the western North Pacific, and on the Amazon continental shelf. In these regions, the $N_2$ internal tides cannot be neglected in the study of sub-mesoscale dynamics. Conversely, the upcoming SWOT data also offer a great opportunity to explore $N_2$ internal tides.

We have examined the possible cross-talk between the $N_2$ and $M_2$ internal tides in our mapping procedure. We map $N_2$ internal tides using two different data sets. The first is the original satellite altimetry SSH data set (Section 2.1). The second is the $M_2$-corrected data set. In other words, the $M_2$ internal tides are predicted using our empirical model and subtracted from the original data. We find that the resulting $N_2$ internal tides from the two data sets are almost the same. The variance of their differences is $<$1% that of the $N_2$ internal tides. Likewise, we map $M_2$ internal tides using both the original and $N_2$-corrected data sets, respectively, and find that the impact of $N_2$ on $M_2$ is negligible. Our analysis reveals that the $N_2$ and $M_2$ internal tides do not crosstalk in our mapping method. It is because the 27-year-long satellite data from 1993 to 2019 are sufficient long to unambiguously separate the $N_2$ and $M_2$ tidal constituents (about 14 minutes apart).

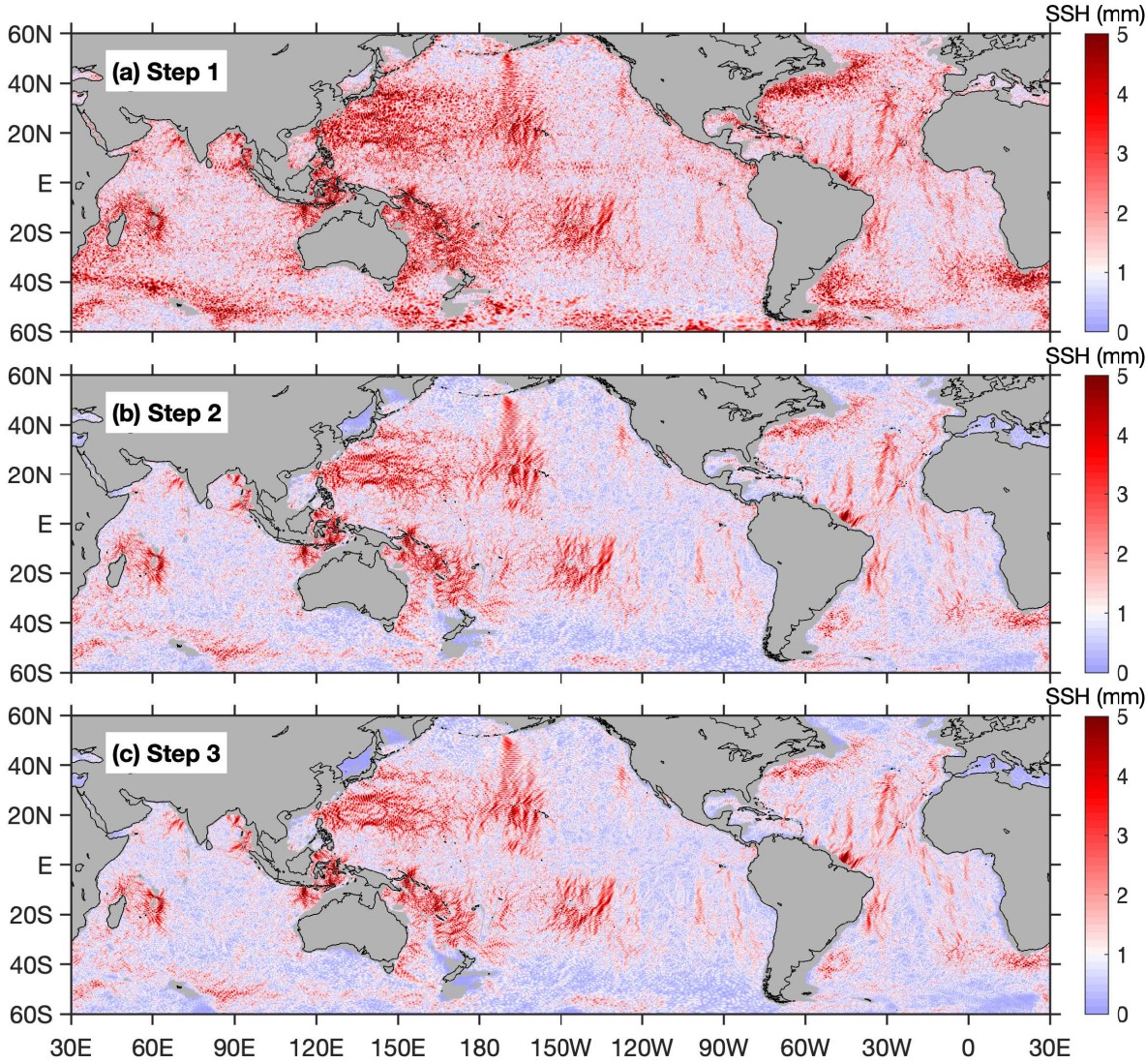

**Figure 3.** The 3-step mapping procedure. (a) Mode-1 $N_2$ internal tides are obtained by the first-round plane wave analysis. (b) Mode-1 $N_2$ internal tides are cleaned by 2D bandpass filtering. (c) Mode-1 $N_2$ internal tides are decomposed by the second-round plane wave analysis. Shown here is the 5-wave superposed amplitude.

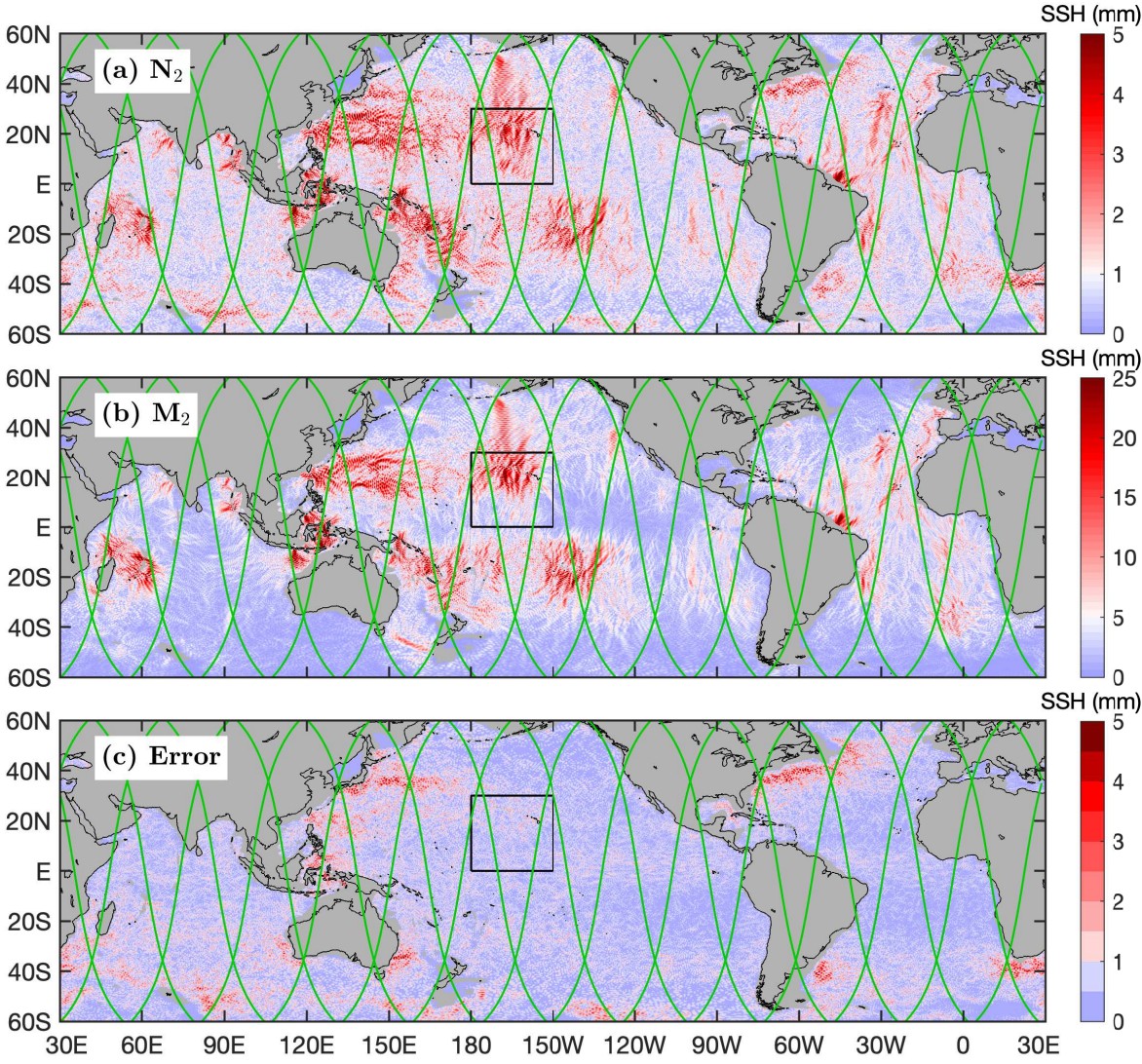

**Figure 4.** Internal tides and model errors. Shown are their SSH amplitudes. (a) $N_2$. (b) $M_2$. $N_2$ and $M_2$ have similar spatial patterns. The $N_2$ amplitudes are about 20% of the $M_2$ amplitudes. (c) Errors. The model errors are obtained by mapping internal tides with a tidal period of 12.6074 hours (Figure 5) using the same satellite altimetry data and the same mapping procedure. Green lines denote SWOT ground tracks in the fast-repeating phase. The mean SSH amplitudes in the black box are given in Figure 5

## 2.5 Model errors

Model errors in our $N_2$ and $M_2$ internal tide models are estimated using background internal tides. In contrast that $N_2$ and $M_2$ internal tides are mapped using tidal periods of 12.6583 and 12.4206 hours, respectively, background internal tides are mapped using the same satellite altimetry data, but for tidal periods between $N_2$ and $M_2$. Specifically, we map 13 sets of background internal tides using 13 different tidal periods that are linearly interpolated between $N_2$ and $M_2$ (Figure 5). The other mapping

parameter, wavenumber (equivalently phase speed), can be obtained using Equation (3). The same strategy has previously been employed to estimate barotropic tide errors. For example, Ray and Susanto (2016) study the fortnightly tidal cycles ($MS_f$ and $M_f$) of tidal mixing using satellite sea surface temperature data. Zaron et al. (2023) study the fortnightly variability of Chl-$a$ using satellite sea surface color data. In both studies, tidal errors are estimated using signals at fake or false tidal frequencies near the real tidal constituents.

We thus obtain 13 background internal tides in the central Pacific (Figure 4c, box). Their regional mean SSH amplitudes in this region are 0.8±0.1 mm, compared to 1.66 and 7.75 mm for $N_2$ and $M_2$ (Figure 5). Note that the SSH amplitudes of the 13 background internal tides are almost same, showing no significant tidal cusps around the $N_2$ or $M_2$ internal tides. In addition, we have calculated the correlation coefficients among these 15 sets of internal tides (including $N_2$ and $M_2$). All correlation coefficients are <0.05, suggesting that these background internal tides are independent with each other and with the $N_2$ and

$M_2$ internal tides. In other words, background internal tides are signals we obtain where there are no tidal constituents. We suggest that the model errors in $N_2$ and $M_2$ can be represented by background internal tides. In this study, we pick one tidal period (12.6074 hours) for a global run to obtain background internal tides (model errors), considering that a global run is time consuming. Figure 4c gives the resulting background internal tides (model errors). It reveals that model errors are large in regions of strong mesoscale motions, because model errors are mainly leaked mesoscale signals. Figure 4 shows that the $N_2$

internal tides are noisier than $M_2$, because the small-amplitude $N_2$ internal tides are easily affected by model errors. On global average, the error variance is about 25% of the $N_2$ variance, and only 1% of the $M_2$ variance.

## 2.6 Model evaluation

Our $N_2$ and $M_2$ internal tide models are evaluated using independent satellite SSH data collected in 2020 and 2021. For each SSH measurement of known time and location, the internal tide signal is predicted using the model under evaluation,

and subtracted from the SSH measurement. Variance reduction is the variance difference before and after the internal tide correction. The variance reductions for all SSH measurements are binned into 2° by 2° boxes. The global map of $N_2$ variance reduction is shown in Figure 6a. The $M_2$ internal tide model is evaluated in the same way and shown in Figures 6b. Note that the colormap ranges for $N_2$ and $M_2$ differ by a factor of 25, that is, the square of the factor of their amplitudes.

In the evaluation, the true $N_2$ internal tides (variance $\sigma_{N2}^2$) in the model will remove the $N_2$ internal tides in the independent

data, leading to positive variance reduction. While the model errors (variance $\sigma_\epsilon^2$) will increase the variance of the independent data, leading to negative variance reduction. Together, we obtain positive variance reduction where $\sigma_{N2}^2 > \sigma_\epsilon^2$, and negative variance reduction where $\sigma_{N2}^2 < \sigma_\epsilon^2$. Figure 6a shows positive variance reduction in the global ocean, suggesting that the true

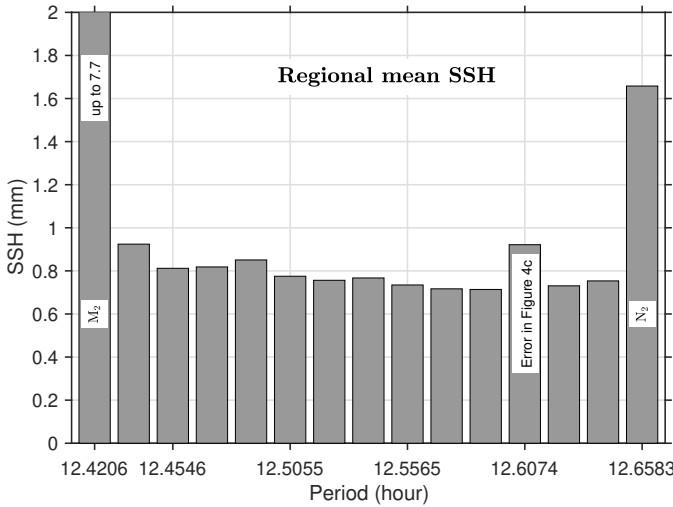

**Figure 5.** Histogram of regional mean SSH amplitudes. In the central Pacific Ocean ($0°$–$30°$N, $180°$–$150°$W), 15 sets of internal tides are determined using the same satellite data and following the same procedure. They are mapped using their respective periods and wavenumbers. Tidal periods of the 13 sets of background internal tides are linearly interpolated between $N_2$ and $M_2$.

$N_2$ internal tides are greater than model errors. In particular, in regions of strong $N_2$ internal tides such as the Hawaiian Ridge and the Amazon continental shelf, patches of positive variance reduction are observed, because the strong $N_2$ internal tides can overcome model errors. While negative variance reduction usually occurs in regions of weak $N_2$ internal tides such as the eastern equatorial Pacific and the Southern Ocean. The regions of strong mesoscale motions are dominated by negative variance reduction, where weak $N_2$ internal tides are overwhelmed by large model errors (Figure 4c). For comparison, Figure 6b shows that the $M_2$ internal tide model causes positive variance reduction throughout the global ocean, except for regions of strong mesoscale motions or strong temporal variation (Zhao, 2021). It is because the strong $M_2$ internal tides are almost always greater than the model errors $\sigma_{M2}^2 > \sigma_\epsilon^2$. In summary, our $N_2$ internal tide model can reduce variance in some regions, although the $N_2$ SSH amplitudes are just a few millimeters.

We further examine the relation between the $N_2$ variance reduction shown in Figure 6a and the variance difference between the $N_2$ model and the model error. Figures 4c and 4d give the $N_2$ model variance and the error variance that are computed from Figures 4a and 4c, respectively. Note that the $N_2$ model variance $\sigma^2$ contains both true $N_2$ internal tides and errors $\sigma^2 = \sigma_{N2}^2 + \sigma_\epsilon^2$. In the condition that the $N_2$ variance is greater $\sigma_{N2}^2 > \sigma_\epsilon^2$, we should have $\sigma^2 > 2\sigma_\epsilon^2$. We thus calculate the variance difference $\sigma^2 - 2\sigma_\epsilon^2$ and show the global map in Figure 6e. To test this relation, we calculate the variance difference $\sigma^2 - m \cdot \sigma_\epsilon^2$ for $m$ ranging from 0.5 to 3.5 with a step of 0.1. For each resulting variance difference map (e.g, Figure 4e), we calculate its correlation coefficient with the $N_2$ variance reduction (Figure 6a). We get the best spatial correlation when the factor $m$ is 2, consistent with our theoretical analysis. Note that all the above analyses are based on the $2°$ by $2°$ binned values. Figure 6f shows the mask region where the $N_2$ model variance is greater than twice the error variance, indicating regions where

the $N_2$ model can be used to make internal tide correction. The mask covers regions of strong $N_2$ and $M_2$ internal tides such as the Hawaiian Ridge, off the California coast, the Amazon continental shelf, the western North Pacific, and the New Caledonia.

We next examine the performance of the $N_2$ and $M_2$ internal tide models in making internal tide correction for SWOT. In Figure 6, the green lines denote the SWOT ground tracks in its daily fast-repeating phase. We interpolate the $N_2$ and $M_2$ variance reductions onto the SWOT ground tracks (neglecting its swath), and calculate the along-track mean variance reductions. For $N_2$, the mean variance reductions in and outside the mask region are 0.73 and $-0.25$ mm$^2$, respectively. The negative variance reduction suggests that the $N_2$ model does not work well outside the mask region. Fortunately, the $N_2$ model can make internal tide correction in the mask region where the $N_2$ internal tides can overcome model errors. The variance reductions caused by the $N_2$ model seem small, but keep in mind that (1) internal tides and sub-mesoscale motions both have millimeter-scale SSH amplitudes, and (2) internal tides are much stronger in their source regions. For $M_2$, the along-track mean variance reductions in and outside the mask region are 25.6 and 2.5 mm$^2$, respectively. They suggest that the $M_2$ model performs well both in and outside the mask region, because the $M_2$ internal tides dominate errors throughout the global ocean.

## 3   Results

### 3.1   Global distribution

Our mode-1 $N_2$ model reveals that $N_2$ internal tides are widespread in the global ocean (Figure 4a). In the Indian Ocean, they are observed in the Arabian Sea, the Bay of Bengal, and the Madagascar-Mascarene region. In the Pacific Ocean, $N_2$ internal tides occur in regions such as the French Polynesian Ridge, the Hawaiian Ridge, the Indonesian seas, the western South Pacific, and the western North Pacific. In the Atlantic Ocean, $N_2$ internal tides appear in regions including the Azores region, the Amazon shelf, the Bay of Biscay, and the Vitoria-Trindade Ridge. Our $M_2$ model reveals that mode-1 $M_2$ internal tides are observed in the same regions (Figure 4b). The $N_2$ and $M_2$ internal tides have similar spatial patterns, but the $N_2$ amplitudes are about 20% of the $M_2$ amplitudes. To further quantify their relation, we give in Figure 7a the scatter plot of the $N_2$ and $M_2$ SSH amplitudes. It shows that the $N_2$ and $M_2$ amplitudes largely follow the diagonal line with a ratio of 5. Their correlation coefficient is 0.69 (R in Matlab function *corrcoef*).

We extract the $N_2$ and $M_2$ barotropic tides from TPXO.8 (Egbert and Erofeeva, 2002) and show them in Figure 8. We find that the $N_2$ and $M_2$ barotropic tides have similar spatial patterns, and that the $N_2$ amplitudes are about 20% of the $M_2$ amplitudes. We examine the relation between the $N_2$ and $M_2$ barotropic tides as well. Figure 7b shows the scatter plot of the $N_2$ and $M_2$ barotropic amplitudes. It shows that $N_2$ and $M_2$ have a very tight relation, with a correlation coefficient of 0.96. Egbert and Ray (2003) show that the $M_2$ and $N_2$ barotropic-to-baroclinic energy conversion maps have similar spatial patterns, and that their amplitudes differ by a factor of 25 (see their Figure 1). The $N_2$ and $M_2$ relation (spatial pattern and amplitude ratio) is the same for both barotropic and baroclinic tides. Because $N_2$ and $M_2$ have close tidal periods (12.6583 and 12.4206 hours), their generations over the same topographic features should be the same (distinguishing their slight differences may improve our understanding of internal tide dynamics in the future). In addition, it is reasonable that the $N_2$ and $M_2$ internal tides have

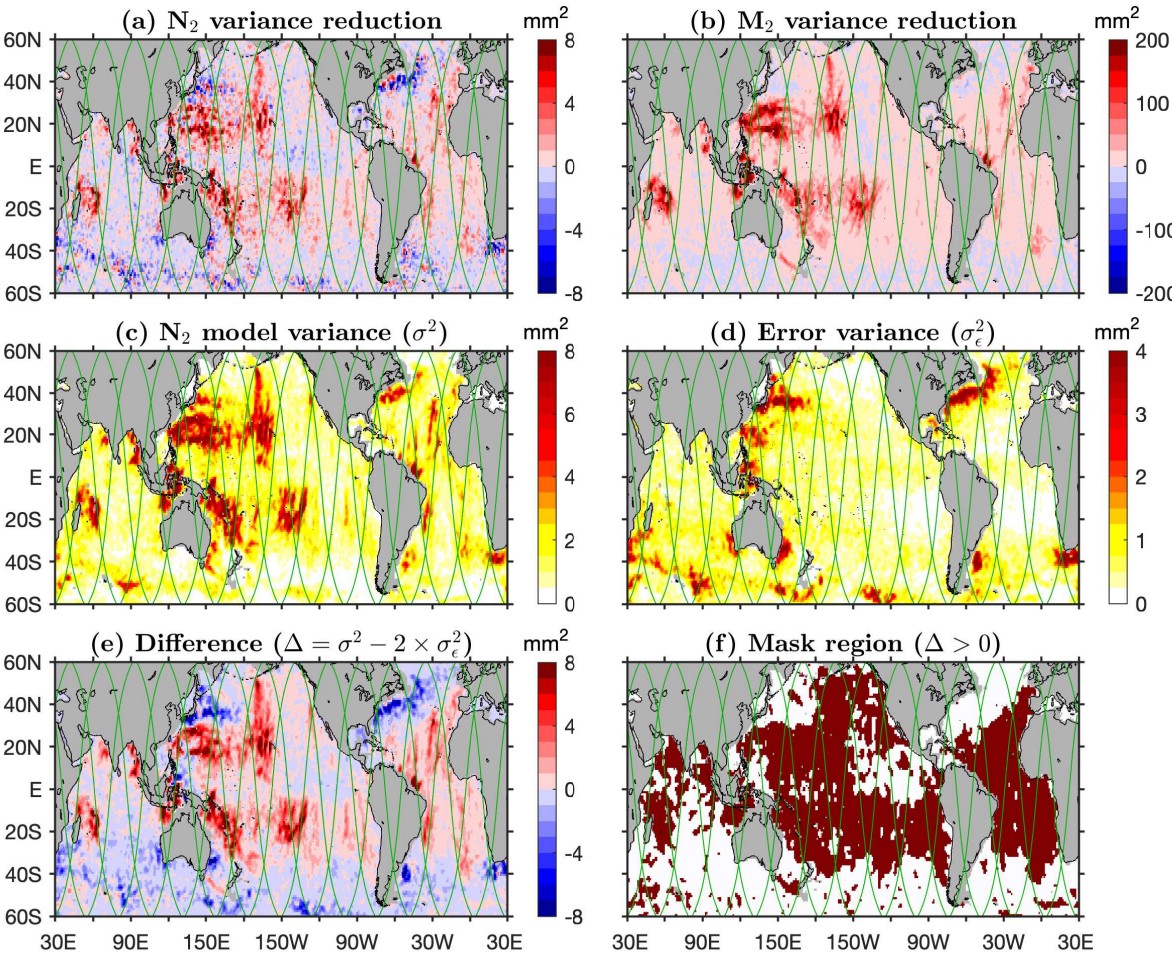

**Figure 6.** Model evaluation. Internal tide models are evaluated using independent altimetry data in 2020 and 2021. Shown are variances or variance reductions binned in $2°$ by $2°$ boxes. (a) $N_2$. (b) $M_2$. (c) Variance of the $N_2$ model. (d) Variance of the model error. (e) Difference between the $N_2$ model variance and twice the error variance. The factor of 2 is chosen to obtain the best spatial correlation between (a) and (e). (f) Mask region where the model variance is greater than twice the error variance, which implies the true $N_2$ variance is greater than the error variance. The mask indicates regions where the $N_2$ model can be used to make internal tide correction. Green lines denote SWOT ground tracks in the fast-repeating phase.

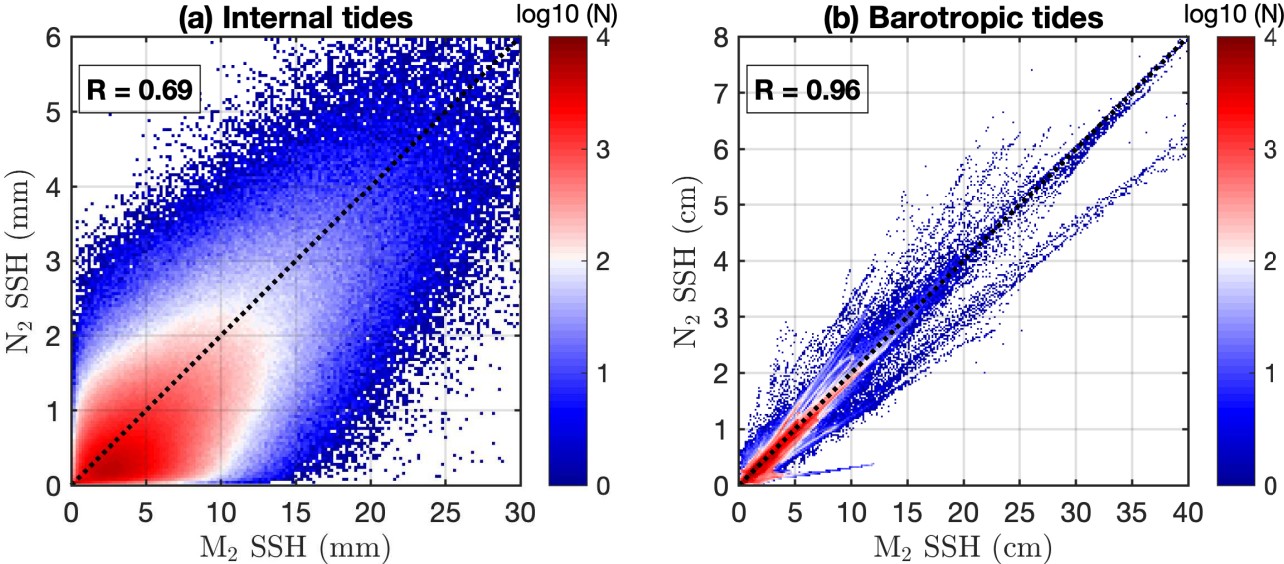

**Figure 7.** Scatter plots of $N_2$ and $M_2$ amplitudes. (a) Internal tides. (b) Barotropic tides. Shown is data density in logarithmic scale. Their correlation coefficients are calculated. For both the barotropic and internal tides, the $N_2$ amplitudes are about 20% of the $M_2$ amplitudes.

a relatively weak relation (Figure 7a), because the long-range propagation of internal tides is affected by an inhomogeneous ocean environment.

## 3.2 Long-range beams

In this section, we study the long-range mode-1 $N_2$ internal tidal beams. We have fitted five mode-1 $N_2$ internal tidal waves at each grid point by plane wave analysis. Taking advantage of the 5-wave fits, we can decompose the $N_2$ internal tide field into the northward (0°–180° anti-clockwise from due east) and southward (180°–360°) components by propagation direction. Each component contains internal tidal waves with propagation directions falling in the given range (Figure 9). The decomposed components clearly show well-defined long-range $N_2$ internal tidal beams, which are characterized by larger amplitudes and cross-beam co-phase lines (not shown here for clarity; see Figures 10 and 11). There are numerous long-range $N_2$ internal tidal beams, which radiate from the strong generation sites mentioned above. For example, northward $N_2$ beams are observed to originate from the French Polynesian Ridge, the Macquarie Ridge, the Amazon shelf, and so on. Southward $N_2$ beams are observed to originate from the Andaman Islands, the Lombok Strait, the Hawaiian Ridge, the French Polynesian Ridge, the Mendocino Ridge, the Azores Islands, among others. Note that the $M_2$ long-range internal tidal beams have been well studied in previous studies (Zhao et al., 2016, Figure 5 therein). To avoid repetition, the $M_2$ internal tidal beams are not shown here. Together, we observe that the $N_2$ and $M_2$ internal tides have similar long-range beams. In this study, we examine two long-range internal tidal beams as examples.

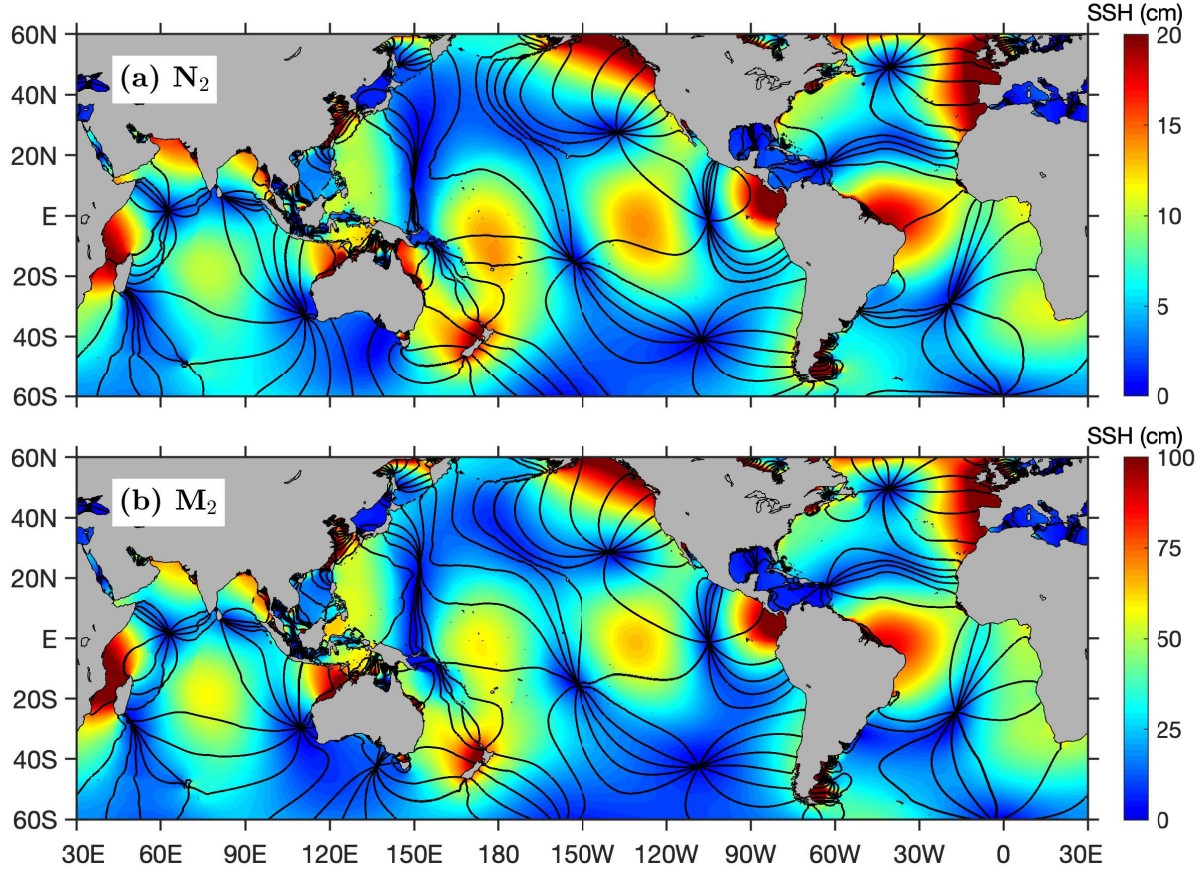

**Figure 8.** $N_2$ and $M_2$ barotropic tides from TPXO.8. Colors show amplitudes. Black lines show co-phase charts at $30°$ intervals. $N_2$ and $M_2$ have similar spatial patterns, and the $N_2$ amplitudes are about 20% of the $M_2$ amplitudes.

First, we examine the southward internal tides from the Amukta Pass, Alaska. The $M_2$ long-range beam from Amukta Pass has been studied recently (Zhao, 2022b). Figure 10a shows the southward $N_2$ internal tides in the central North Pacific (Figure 9b, blue box). For comparison, the southward $M_2$ internal tides are shown in Figure 10b. Both tidal constituents can travel from the Aleutian island chain to the Hawaiian Ridge over 3,000 km away. Their propagation directions are about $-78°$ from due east. The black lines shows the $0°$ and $180°$ co-phase charts. Figure 10c shows their phase difference, which increases with propagation, because $N_2$ internal tides travel faster than $M_2$ internal tides according to Equation (3). In the propagation, their phase difference increases with propagation. Along the dashed line from source ($52.6°$N, $189°$E) to far field ($26°$N, $195°$E), their phase difference increases from $65°$ to $305°$. The overall phase change is $240°$. It takes about 18 tidal cycles for the $N_2$ and $M_2$ internal tides to travel along the path.

Figure 11 shows southward internals tides in the region off the California Coast (Figure 9b, cyan box). This region is chosen for a detailed investigation, because it contains one site for the SWOT calibration/validation field campaign. The green lines

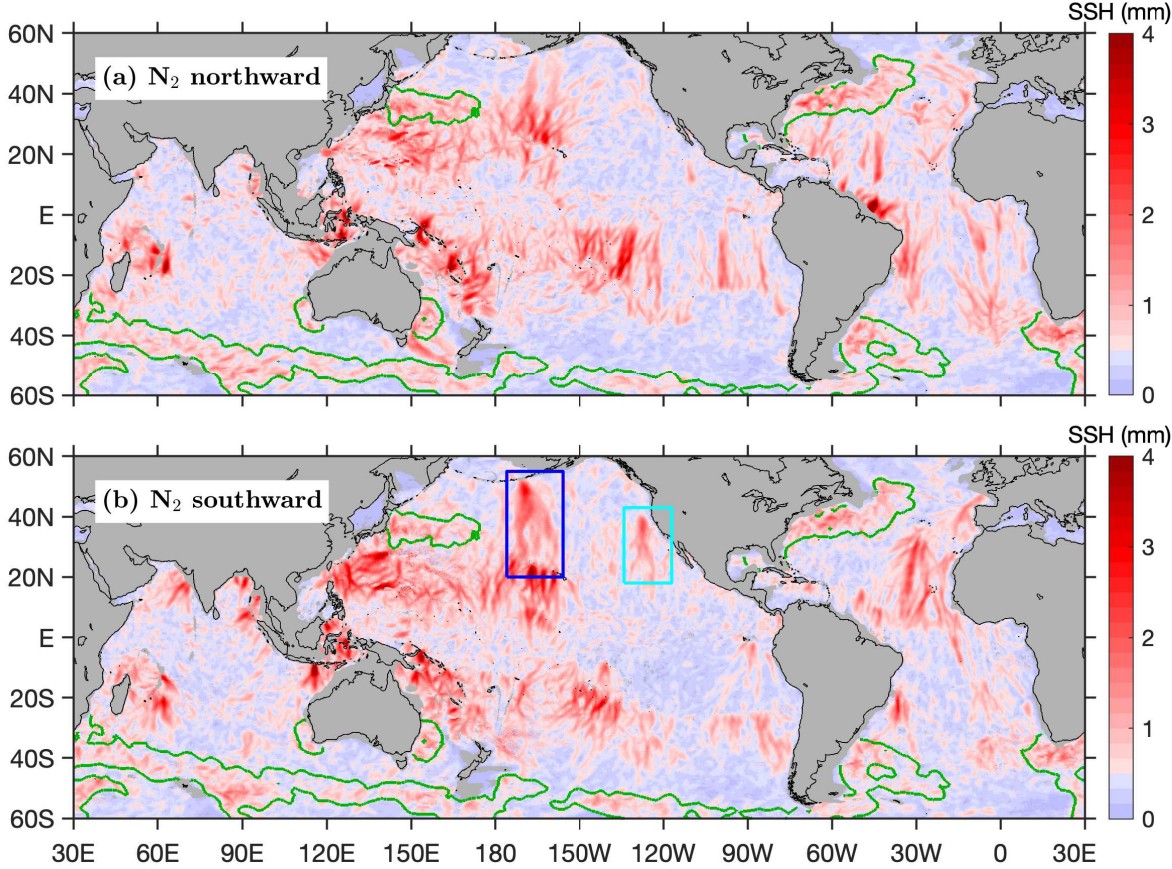

**Figure 9.** Decomposed $N_2$ internal tide fields. (a) Northward component (directional range: $0°$–$180°$ anti-clockwise from due east). (b) Southward component (directional range: $180°$–$360°$). Internal tides in shallow waters ($<1000$ m) are discarded. Green contours indicate regions of strong mesoscale motions, where the $N_2$ internal tides are overwhelmed by errors (see Figure 4e).

in this figure indicate the SWOT swaths in its fast-repeating phase (Wang et al., 2022). The cross-over region of the ascending and descending swaths is the SWOT calibration/validation site. This region is dominated by the southward internal tides from the Mendocino Ridge. Note that this region is also affected by internal tides in other propagation directions (Zhao et al., 2019). Additionally, there are southwestward internal tides from the Monterey Bay. The two internal tidal beams intersect around the SWOT campaign site. As explained earlier, our $N_2$ model can make internal tide correction for SWOT. Figure 11 shows that

$N_2$ and $M_2$ internal tides are very similar, although the $N_2$ fluxes are much weaker. Both $N_2$ and $M_2$ beams can be tracked from $40°$N to $20°$N for $>2000$ km. They both bifurcate around $32°$N near Fieberling Seamounts ($32.5°$N, $232.3°$E) for unknown reasons. The dashed line delineates the beam from $40.3°$N to $22°$N along $128°$W. This line is about 2000 km long. Along this line, the $N_2$ and $M_2$ phase difference increases from $40°$ to $160°$ over about 14 $M_2$ or $N_2$ tidal cycles.

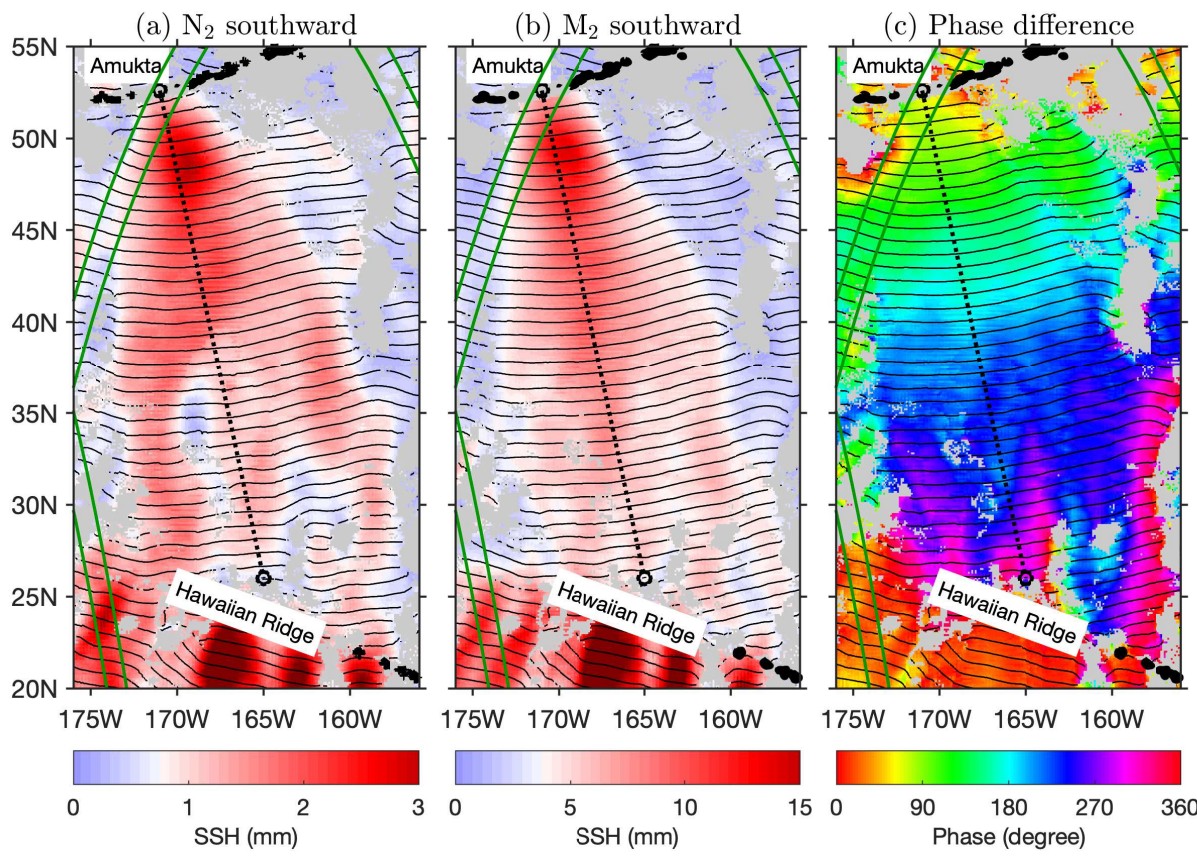

**Figure 10.** Southward (180°–360°) internal tides from Amukta Pass, Alaska. (a) $N_2$ internal tides. (b) $M_2$ internal tides. Black contours show the 0° and 180° co-phase lines. Intervals between two neighboring co-phase lines are half wavelengths. (c) Phase difference between $N_2$ and $M_2$. Black contours show the $M_2$ co-phase lines as in (b). Weak internal tides are discarded ($M_2 <1$ mm; $N_2 <0.2$ mm). Both $N_2$ and $M_2$ internal tides can propagate over 3000 km from the Aleutian Ridge to the Hawaiian Ridge. Along the 3000-km-long path (dashed lines), their phase difference increases from 65° to 305°. Green lines denote the 120-km-wide SWOT swaths in the fast-repeating phase.

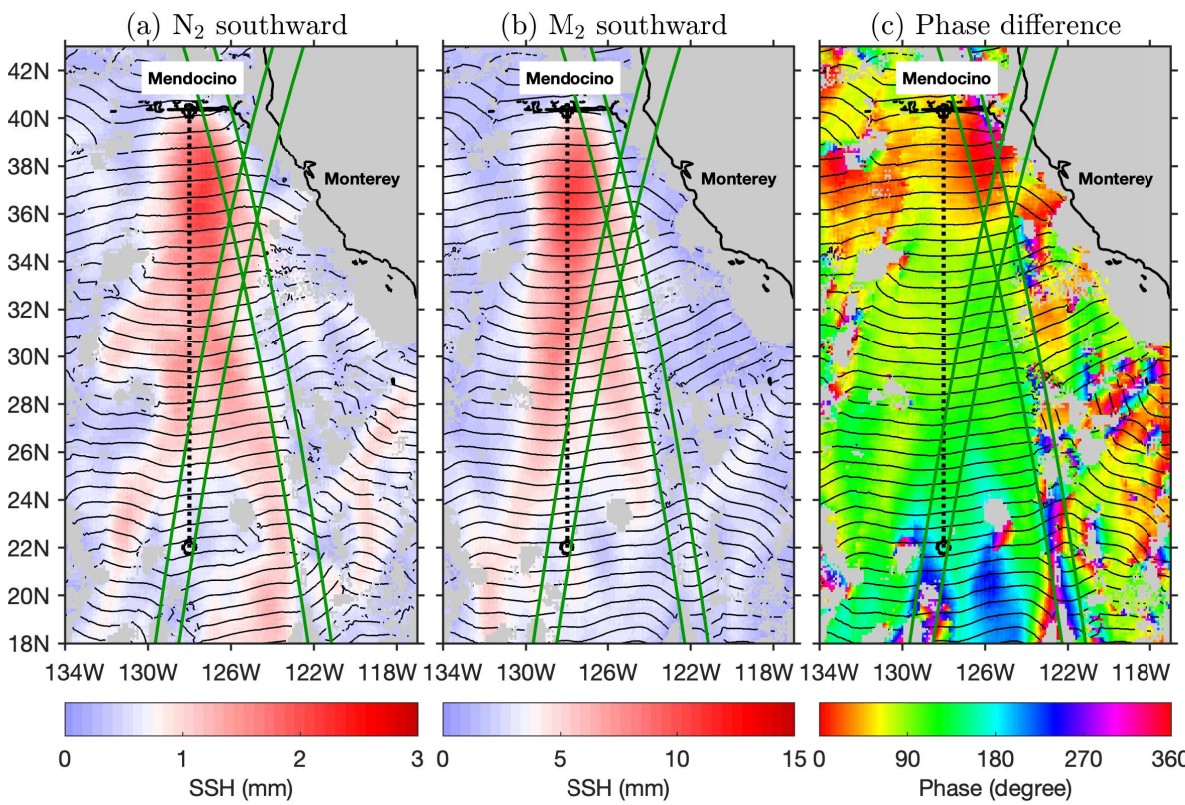

**Figure 11.** Southward ($180°$–$360°$) internal tides off the U.S. West Coast. (a) $N_2$ internal tides. (b) $M_2$ internal tides. Colors show SSH amplitudes. Black contours indicate the $0°$ and $180°$ co-phase lines. Southward internal tides are mainly from the Mendocino Ridge. Weak internal tides are discarded ($M_2$ <1 mm; $N_2$ <0.2 mm). Internal tides in shallow waters (<1000 m) are discarded. (c) Phase difference between $N_2$ and $M_2$. Green lines denote two 120-km-wide SWOT swaths in the fast-repeating phase.

### 3.3 Energy and energy flux

We calculate the depth-integrated energy flux of mode-1 $N_2$ internal tide from their SSH amplitudes and a transfer function ($F_n$). The transfer function is calculated using the WOA18 climatological hydrography and the Sturm-Liouville equation (Zhao and Alford, 2009; Zhao et al., 2016). The same calculation method has also been derived by Geoffroy and Nycander (2022). In this study, we follow our method (previously for mode-1 $M_2$ internal tides) to obtain the transfer function for mode-1 $N_2$ internal tides. It is a function of ocean depth, tidal frequency, mode number, latitude, and stratification. The transfer functions

for $N_2$ and $M_2$ are very close, because their tidal periods are close. At each grid point, we thus obtain five energy fluxes for the five internal tidal waves following $F = \frac{1}{2}F_n A^2$, where $A$ is the SSH amplitude. The vector sum of the five energy fluxes gives the final energy flux at this site. In this study, we compare the $N_2$ and $M_2$ internal tide energy fluxes in two regions. An interested reader can examine other ocean regions. We show that their energy fluxes have similar spatial patterns. The

results show that the mode-1 $N_2$ internal tides can be observed by satellite altimetry, although their are much weaker than the $M_2$ internal tides. Following the same procedure, we have computed the depth-integrated internal tide energies from SSH amplitudes. The globally integrated area-weighted energies for the $N_2$ and $M_2$ internal tides are 1.8 and 30.9 PJ, respectively. The $N_2$-to-$M_2$ ratio is about 5.8%, larger than the theoretical value of 4%, because $N_2$ contains larger error variance. As explained earlier, the error variance is about 25% of the $N_2$ variance, but only 1% of the $M_2$ variance.

Figure 12 shows the $N_2$ and $M_2$ energy fluxes in the western South Pacific. In this study, it is trimmed to 30°S–0°, 145°E–125°W. Colors show flux magnitudes, and black arrows show flux vectors. This region is chosen because (1) it features various topographic obstacles such as mid-ocean ridge and island chain, and (2) the New Caledonia region is one site for SWOT calibration/validation field experiment (Bendinger et al., 2023). There are numerous $N_2$ and $M_2$ internal tidal beams in this region. They are dominantly generated over topographic features. For example, $N_2$ and $M_2$ internal tidal beams radiate from many straits surrounding the Coral Sea (Tchilibou et al., 2020). The internal tidal beams can be in any horizontal propagation direction. From the French Polynesian Ridge, internal tides mainly propagate southward and northward. From the Kermadec Arc and the New Caledonia, the outgoing internal tidal beam usually travel eastward or westward. The energy fluxes of $N_2$ and $M_2$ internal tides have similar spatial patterns. Figure 12 shows seven SWOT swaths in this region (green lines). Among them, the two swaths in the New Caledonia region (black box) overlap with strong $N_2$ internal tides whose contribution cannot be neglected. In addition, the two swaths are cross the French Polynesian Ridge, where one should pay an attention to $N_2$ and $M_2$ internal tides in the study of mesoscale and sub-mesoscale processes.

Figure 13 shows the $N_2$ and $M_2$ energy fluxes in the North Atlantic Ocean (2°S–53°N, 58°W–3°W). Figure 13 is in the same format as Figure 12. Internal tides in this region have attracted much attention in recent years (Vic et al., 2018; Köhler et al., 2019; Löb et al., 2020). In particular, internal tides on the Amazon continental shelf have been intensively studied recently, partly because of the co-existence of internal tides and internal solitary waves (Egbert and Erofeeva, 2021; Tchilibou et al., 2022; Assene et al., 2023). Our satellite observation reveals that strong $N_2$ and $M_2$ internal tides occur around notable topographic features including the Mid-Atlantic Ridge, the Amazon continental shelf, the Azores region, the Bay of Biscay, Canary Islands, and the Cape Verde islands. The longest internal tidal beams for both $N_2$ and $M_2$ are the southward internal tidal beams from the Azores (Zhao, 2016; Köhler et al., 2019). The two beams can be tracked over 2000 km. In this region, there are four SWOT swaths in its fast-repeating phase, which overlap remarkable $N_2$ and $M_2$ internal tidal beams.

## 4  Summary

In this study, we constructed empirical models for mode-1 $N_2$ and $M_2$ internal tides from satellite altimetry. Among them, $N_2$ is the larger lunar elliptical semidiurnal constituent and the fifth largest oceanic tidal constituent. It is induced by the Moon's elliptical orbit. Its amplitudes are about 20% of the $M_2$ amplitudes. The mode-1 $N_2$ internal tides have sub-centimeter-scale SSH amplitudes. We can extract weak $N_2$ internal tides, because we use a larger altimetry data set and a newly-developed mapping procedure. First, we use the multiyear multisatellite altimetry data from 1993 to 2019. The combined data are about 100 satellite-years long, which can significantly suppress nontidal errors. Second, we extract mode-1 $N_2$ internal tides by a

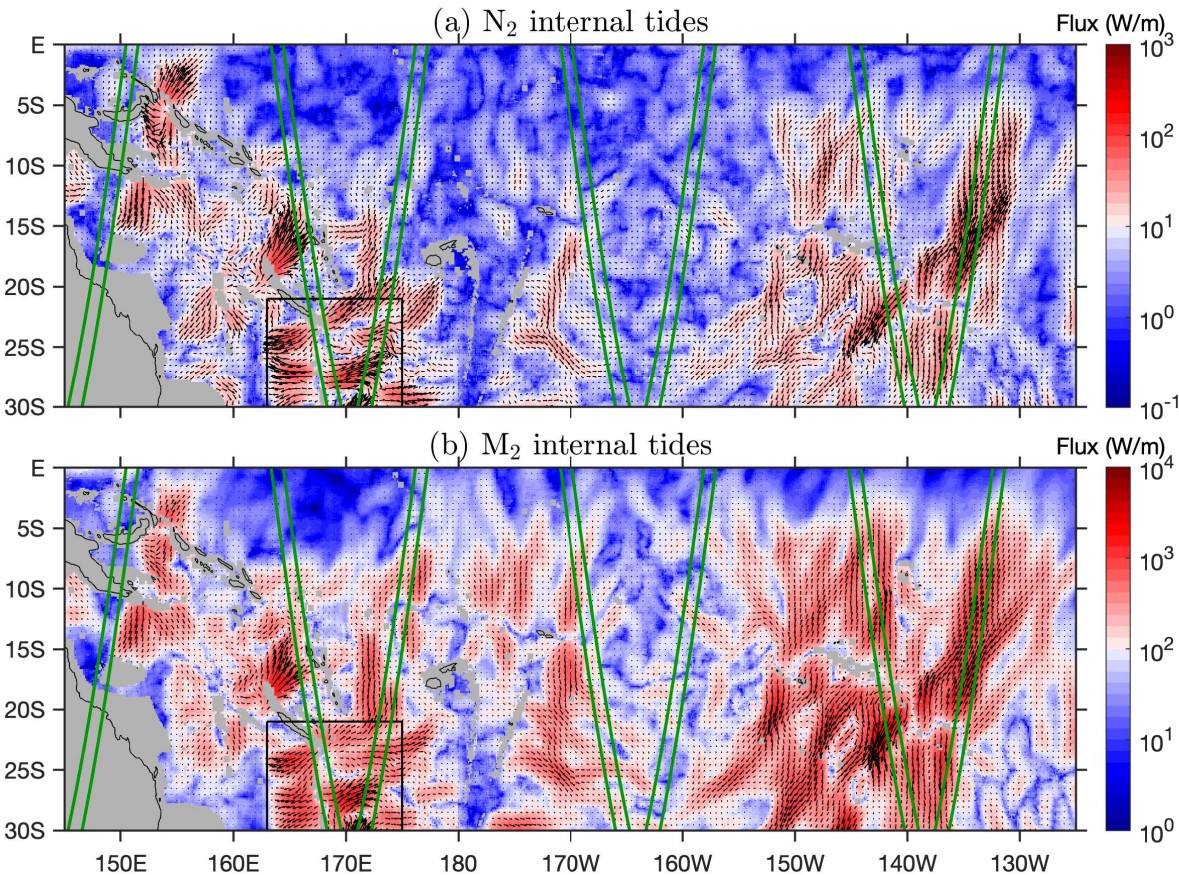

**Figure 12.** Internal tide energy fluxes in the western South Pacific. (a) $N_2$ internal tides. (b) $M_2$ internal tides. Colors show flux magnitudes and black arrows show flux vectors. Internal tides in shallow waters (<1000 m) are discarded (gray shading). Green lines indicate SWOT swaths in the fast-repeating phase.

three-step mapping procedure, which cleans internal tides using known frequency and wavenumbers of the target internal tide. In consequence, satellite altimetry can observe mode-1 $N_2$ internal tides with millimeter-scale SSH amplitudes. Our $N_2$ internal tide model is still noisy. Future improvements can be made with more and more satellite altimetry data becoming available.

We estimated errors in the $N_2$ and $M_2$ internal tide models using background internal tides. Specifically, background internal tides are mapped using the same altimetry data but for tidal periods between $N_2$ and $M_2$. In this study, we construct a global map of model errors using a tidal period of 12.6074 hours ($N_2$ minus 3 minutes). The model errors are usually <1 mm in the global ocean, with the global mean error being about 0.7 mm. Large errors usually occur in regions of strong mesoscale motions, since the model errors mainly come from the leaked mesoscale signals. On global average, the error variance is about
25% of the $N_2$ model variance, but only 1% of the $M_2$ model variance.

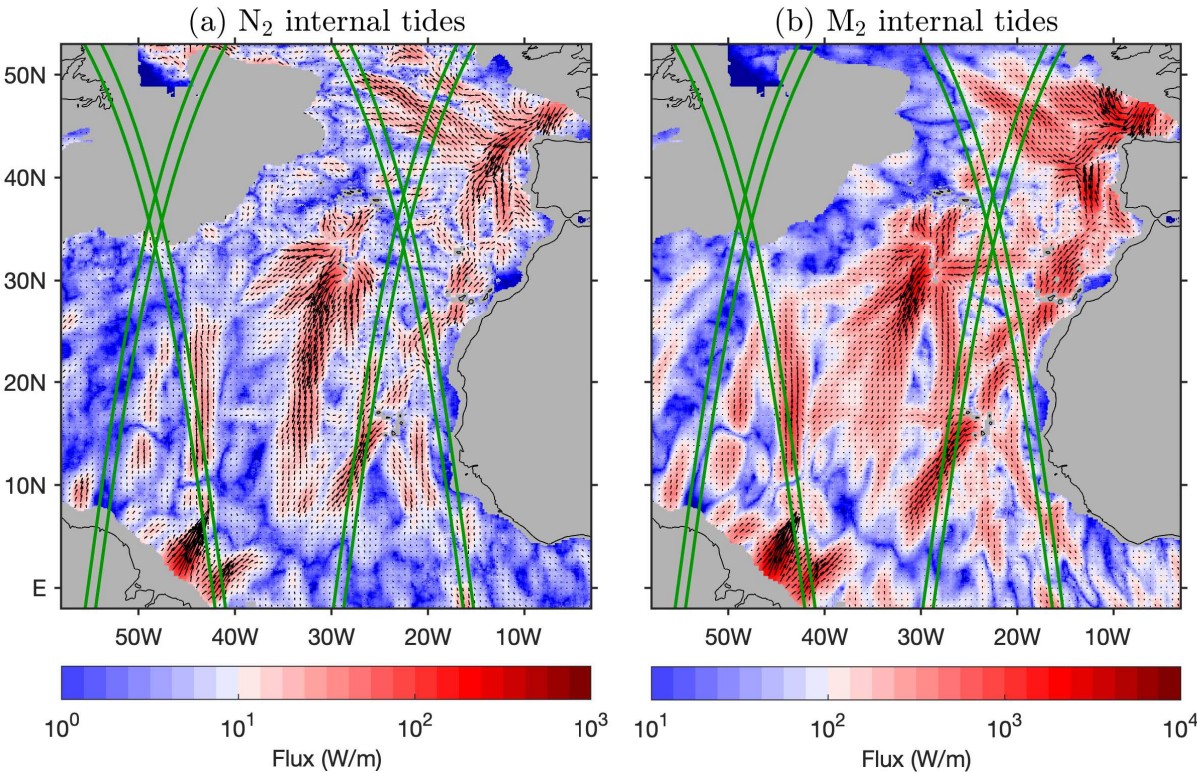

**Figure 13.** As in Figure 12 but for internal tide energy flux in the North Atlantic Ocean.

Our satellite observations revealed some basic features of the global $N_2$ internal tides. We found that the $N_2$ and $M_2$ internal tides have similar spatial patterns, and that the $N_2$ amplitudes are about 20% of the $M_2$ amplitudes. Both features are determined by their barotropic counterparts. We found that both $N_2$ and $M_2$ internal tides can propagate hundreds to thousands of kilometers in the open ocean, but at different phase speeds. We examined regional $N_2$ internal tides and revealed rich information on their generation and propagation. We suggest that including $N_2$ internal tides can better simulate the temporal variation of internal tide energetics with the lunar elliptical orbit.

Our $N_2$ and $M_2$ internal tide models have been evaluated using independent altimetry data in 2020 and 2021. The $M_2$ model can cause variance reduction throughout the global ocean, because the $M_2$ internal tides dominate the model errors. In contrast, the $N_2$ model can cause variance reduction in regions of strong $N_2$ internal tides where they can overcome errors. We found that the $N_2$ model performs well in regions where the $N_2$ model variance is greater than twice the error variance, which means that the true $N_2$ variance is greater than the error variance. We showed that the $N_2$ and $M_2$ models work well in the mask region along the SWOT fast-repeating tracks, which suggests that they can make internal tide correction for SWOT.

Last but not least, we demonstrated that our mapping technique can construct a reliable mode-1 $N_2$ internal tide model using 100 satellite-years of altimetry data. We have applied our mapping technique to the first baroclinic mode of other minor tidal

constituents and higher baroclinic mode of other major tidal constituents, and obtained clear internal tide signals. We have tried mapping mode-2 $N_2$ internal tides around the Hawaiian Ridge (185–205°E, 18–28°N). However, the resulting model is noisy, as expected. In this region, the mean amplitude of mode-1 $N_2$ internal tides is about 2.5 mm. The mean mode-2 $N_2$ amplitude is estimated to be 1 mm, using a ratio of 2.5 from mode-1 and -2 $M_2$ internal tides. The ∼1-mm mode-2 $N_2$ internal tides cannot overcome the ∼0.7-mm noise. It is expected that the low-noise SWOT data along 120-km-wide swaths will improve

the observation of minor tidal constituents and higher baroclinic modes.

*Data availability.* The satellite altimetry along-track data are from the Copernicus Marine Service (https://doi.org/10.48670/moi-00146). The satellite altimetry gridded data are from the Copernicus Marine Service (https://doi.org/10.48670/moi-00148). The SWOT orbit data are from the AVISO website (https://www.aviso.altimetry.fr/en/missions/current-missions/swot). The World Ocean Atlas 2018 is produced and made available by NOAA National Oceanographic Data Center (https://www.nodc.noaa.gov/OC5/woa18/).

The mode-1 $N_2$ internal tide model developed in this study is freely available (https://doi.org/10.6084/m9.figshare.23243633.v1)

*Author contributions.* This paper is completed by the sole author.

*Competing interests.* The author has declared there are no competing interests.

*Acknowledgements.* This study was funded by the National Aeronautics and Space Administration (NASA) via projects NNX17AH57G and 80NSSC18K0771.

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
