# Peer review of "Mode-1 N2 internal tides observed by satellite altimetry"

_EGUsphere, 2022_

## Referee Comment (RC1)

Review of manuscript "Mode-1 N2 internal tides observed by satellite altimetry" by Zhongxiang Zhao.

Manuscript reviewed by Clément Vic on 14th November 2022

The manuscript reports on a method that allows to map the global mode-1 N2 internal tides from sea surface height (SSH) measured by satellite altimetry. Results are presented and discussed in light of a similar product for the well-described mode-1 M2 internal tides. The model's predictive skill is tested vis-à-vis of an independent global time series of SSH, and it proves to be good. The manuscript is clearly written and well presented, and the science looks robust to me. The scientific interest of mapping internal tides is emphasized in the context of the new SWOT satellite mission. I recommend this manuscript for publication after some minor revisions that I hope can help to clarify some points.

Minor comments:
- Line 6 and elsewhere: "1993 through 2019" replace "through" with "to"?
- Line 10: "Both features mimic their barotropic counterparts" is unclear to me. This could be rephrased and developed a bit, including considerations on the energy conversion etc.
- Line 20: the acronym should be expanded first and put into parentheses afterward, i.e., "Surface Water and Ocean Topography (SWOT)"
- Line 20: The SWOT mission has not been designed to strictly study the submesoscales only I think, but to have a wider picture of entangled motions and waves that have a signature in SSH.
- Line 27: remove "(Section 2)" and stick it to where the plan is announced?
- Line 31: I think it would be useful for the reader to be introduced to the semidiurnal tide components in general, before getting into the details of N2 and L2.
- Line 37: insert "see their Figure 4" within the previous parentheses?
- Line 39: "can better parameterize" is not very clear. I think you mean more than just parameterize, i.e., increase our understanding?
- Line 70: why is the fitting window size not varying geographically? I thought it would take into account mode-1 wavelength.
- Line 75: add "horizontal" before "wavenumber"
- Line 98 (and perhaps before): It would be good to stress out that only the coherent internal tides are mapped.
- Line 112: rephrase "they have same generations over rough topography" to be more specific (amplitude, phase, etc.)
- Line 114: add "energy" before "conversion"
- Line 115: "in the causative chain" is unclear. If I understood correctly, you mean that the similar M2 and N2 barotropic tides generate similar baroclinic tides and that the ratio between barotropic tides is also observed in baroclinic tides.
- Line 120: I assume 0 degree is East? It should be mentioned.
- Line 137: typo? "prone to" be?
- Line 146: remove "percentage"
- Line 157: I do not understand the sentence "Thus, …"

- Figure 4 and line 173: I think it would be good to give globally integrated numbers for all terms, and perhaps the integrated contribution of all negative and positive terms to make sure that the variance is indeed reduced.
- Line 198: add "mode-1"
- Line 201: I do not understand how one can do "parameterizing the temporal variation" with the elements provided. It could be developed.
- Line 208: typo (sub and not sum). And perhaps rephrase into something that reminds the reader on the fast (waves) and slow (balanced motions) manifolds?

---

## Author Comment (AC1)

The manuscript reports on a method that allows to map the global mode-1 $N_2$ internal tides from sea surface height (SSH) measured by satellite altimetry. Results are presented and discussed in light of a similar product for the well-described mode-1 $M_2$ internal tides. The model's predictive skill is tested vis-à-vis of an independent global time series of SSH, and it proves to be good. The manuscript is clearly written and well presented, and the science looks robust to me. The scientific interest of mapping internal tides is emphasized in the context of the new SWOT satellite mission. I recommend this manuscript for publication after some minor revisions that I hope can help to clarify some points.

Thank you very much for your time and suggestions.

Minor comments:

- Line 6 and elsewhere: "1993 through 2019" replace "through" with "to"?

Changed as suggested.

- Line 10: "Both features mimic their barotropic counterparts" is unclear to me. This could be rephrased and developed a bit, including considerations on the energy conversion etc.

This expression is removed.

- Line 20: the acronym should be expanded first and put into parentheses afterward, i.e., "Surface Water and Ocean Topography (SWOT)"

Changed as suggested.

- Line 20: The SWOT mission has not been designed to strictly study the submesoscale only I think, but to have a wider picture of entangled motions and waves that have a signature in SSH.

Changed.

- Line 27: remove "(Section 2)" and stick it to where the plan is announced?

Removed as suggested.

- Line 31: I think it would be useful for the reader to be introduced to the semidiurnal tide components in general, before getting into the details of N2 and L2.

Thank you. But we do not how much the reader knows about tide.

- Line 37: insert "see their Figure 4" within the previous parentheses?

Changed as suggested.

- Line 39: "can better parameterize" is not very clear. I think you mean more than just parameterize, i.e., increase our understanding?

"parameterize" is replaced with "describe"

- Line 70: why is the fitting window size not varying geographically? I thought it would take into account mode-1 wavelength.

Good point. But it does not change much. See my recent paper (Zhao 2022 JPO).

- Line 75: add "horizontal" before "wavenumber"

Changed. 'Horizontal' is added in several places.

- Line 98 (and perhaps before): It would be good to stress out that only the coherent internal tides are mapped.

One sentence is added to admit this limitation.

- Line 112: rephrase "they have same generations over rough topography" to be more specific (amplitude, phase, etc.)

Dropped in revision.

- Line 114: add "energy" before "conversion"

Changed as suggested.

- Line 115: "in the causative chain" is unclear. If I understood correctly, you mean that the similar M2 and N2 barotropic tides generate similar baroclinic tides and that the ratio between barotropic tides is also observed in baroclinic tides.

"In the causative chain" is removed.

- Line 120: I assume 0 degree is East? It should be mentioned.

Yes. We follow standard polar coordinates.

- Line 137: typo? "prone to" be?

Fixed.

- Line 146: remove "percentage"

Removed as suggested.

- Line 157: I do not understand the sentence "Thus, …"

This sentence is dropped.

- Figure 4 and line 173: I think it would be good to give globally integrated numbers for all terms, and perhaps the integrated contribution of all negative and positive terms to make sure that the variance is indeed reduced.

Figure 4 (now Figure 6) is improved. I feel a global integration is not necessary.

- Line 198: add "mode-1"

Changed.

- Line 201: I do not understand how one can do "parameterizing the temporal variation" with the elements provided. It could be developed.

Changed to 'better description of the temporal variation'

- Line 208: typo (sub and not sum). And perhaps rephrase into something that reminds the reader on the fast (waves) and slow (balanced motions) manifolds?

Fixed.

---

## Author Comment (AC2)

This paper discusses the author's efforts to map the baroclinic N2 tide from satellite altimeter data collected from 1993 to 2019. The newly-derived maps are used to predict tides for missions in the 2020-2021 time period, and evaluate the quality of the maps. The mapping methodology has been presented in the author's previous papers. Mapping of baroclinic N2 was previously attempted by Dushaw (2015), and the work in this paper shows that additional data collected since that time, in combination with the author's analysis technique, makes it possible to map the N2 tide with more precision. While the results in this paper may be a step forward, I don't believe that it provides enough new material or insight to justify publication at this time. Although, with some added analysis, the paper could stand on its own and make a worthwhile contribution to the literature.

Thank you for your time and suggestions. My initial plan is to publish a short note. Now I change it to a long paper with 13 figures (vs 4 figures). In the revised manuscript, I present more details and more scientific insights.

Major Comments:
(1) The paper's qualitative observations about $N_2$ in relation to $M_2$ are completely plausible and expected, as the author points out.

Now the relation between $M_2$ and $N_2$ tides is quantitively studied. Figure 7 shows scatter plots of their amplitudes for both barotropic and baroclinic modes. Their correlation coefficients are given.

I don't understand the significance of the analysis of wave dispersion (section 3.3), though, since it seems to just verify the properties of the wave fields which are already assumed by the analysis technique. I would be surprised if there is anything of significance to say about the dynamics of N2 in contrast to M2, but maybe the author can convince me otherwise.

This analysis is removed. Now the phase difference of $M_2$ and $N_2$ internal tides are simply examined along two long-range beams.

(2) At line 48, the author states, "the resulting N2 internal tides still have considerable errors ... and work as a useful internal tide model," but I believe this is unproven. The qualitative discussion of the errors, from lines 170 to 183, may be correct, but there are certainly plenty of locations in Fig 4a where N2 is large-amplitude, but it exhibits negative explained variance (e.g., in the Western Pacific and Philippines Sea). If the discussion of noise or error level of the tide model could be made precise, then it might provide an objective means of deciding where the model can or cannot be a "useful internal tide model." Otherwise, it is just guesswork whether this could be used to provide a useful correction for the SWOT mission.

In the revised manuscript, the model errors are objectively estimated and shown in Figures 4c and 5. The model evaluation is improved, and Figure 6 clearly shows regions of positive or negative variance reduction.

Minor Comments:
l3: "technique" -> "techniques"
Fixed.

l57-58: Some of these missions have not been used previously for the estimation of N2 tides. What are the alias periods of N2 for the following missions: Cryosat-2, Haiyang-2A/2B, and Sentinel-3A/B? The Copernicus web site mentions that the products used are "tailored for data assimilation". It would be useful to explain what this involves.

Figure 2 is added to check the tidal aliasing issue. I check the histogram of the phase lags of the SSH measurements in one typical fitting window with respect to one $N_2$ ($M_2$) tidal cycle. The result shows that the SSH data evenly distribute over one $M_2$ cycle. Their distribution over one $N_2$ cycle is a little bumpy, which may cause larger errors in the $N_2$ model.

The satellite along-track data I used in this study are along-track unfiltered data. Thus, ALL internal tide signals are retained. My results (Figure 4) clearly show that the empirical $N_2$ model is informative, though imperfect. In this regards, the SWOT data can help.

l80-81: "orthogonal equation" ? Please use correct terminology

Changed to "*the Sturm-Liouville equation*."

l86: I am curious why this bandwidth was selected and what are the implications?

Figure S1 below shows my 2D bandpass filter and the bandwidth. The bandwidth (upper cutoff and lower cutoff wavenumbers) is indicated by the two concentrical circles in (b). They are empirically chosen to be [0.8 1.25] times the local wavenumber $k$. It is because the 2D spectrum (b) shows outstanding peaks near $k$. If the bandwidth is too wide (say [0.5 1.5]), the output field will contain more background errors. If the bandwidth is too narrow (say [0.9 1.25]), the output field will lose real internal tide signals.

l137: typo: "beams are prone to affected" Also -- the beams are not affected by the measurement noise; the estimates for these features are affected by the noise.

Thank you. Changed. Later it is removed.

Section 3.3: I don't really understand the significance of this section. The frequencies of the tides are given, and the wavenumbers are assumed given by the dispersion relation. Thus, the observations of wave dispersion reported here are a consequence of the assumptions of the analysis method, aren't they?

This analysis is removed. The phase difference is studied along two long-range beams.

Section 3.4: Figure 4a shows that the proposed model of N2 does explain SLA variance in some regions and not others, as the text states. I think it would be important to provide an error estimate for the N2 tide. How would someone know where the correction should be applied and where it should not be applied? The qualitative discussion in the text mentions the amplitude of N2 in relation to the model error, sigma_epsilon, but this quantity is not estimated, even though it appears in mathematical inequalities.

In the revised manuscript, the model errors are objectively estimated and shown in Figures 4c and 5. The model evaluation is improved. My new Figure 6 clearly shows regions of positive or negative variance reduction.

l205: "sum-mesoscale" ?

Fixed.

Figures: I would like to see the figures much larger; I cannot make out much detail in the global maps in the sizes they are presented.

In the revised manuscript, all figures are plotted large.

[Figure]

Figure S1. Horizontal 2D bandpass filter. Shown is a typical 850 km by 850 km window. (a) Snapshot $N_2$ internal tide field. It is obtained in the first-round plane wave analysis. (b) 2D wavenumber spectrum of (a). The two concentric circles indicate the upper and lower cutoff wavenumbers of the 2D filter. They are 0.8 and 1.25 times the mean wavenumber in this window. (c) Snapshot output $N_2$ internal tide field. (d) Difference between the input and output fields.

---

## Author Comment (AC3)

Replies to RC #3

This manuscript presents a global map of the sea-surface height (SSH) signals of the $N_2$ internal tides observed by satellite altimetry. The topic is important for mapping the tidal energy available for diapycnal mixing and also for reducing the tidal signals in SSH data in the studies of mesoscale and sub-mesoscale phenomena. Although I do not understand the technical details, the presented maps of the $N_2$ internal tides seem all reasonable.

Thank you for your time and suggestions.

Minor Comments:

The explanation about the method for extracting the N2 internal tides (section 2.2) is hard to understand. I would suggest the authors rewrite this part more to the point, or omit it. Right now, the readers who are not very familiar with this method cannot judge whether this method is really suitable for mapping the N2 internal tides.

In the revised manuscript, I rewrite my methods and give more details. Now Figures 1–5 give more information on data, tidal aliasing, mapping methods, and model errors.

A few specific examples of the above comment are:

1. Why are "five" plane-waves needed to fit the N2 internal tides?

The mode-1 $N_2$ internal tide field contains long-range internal tides originating from various source regions. We thus should fit multiple internal tidal waves at each grid point. In this study, I fit five mode-1 $N_2$ internal tidal waves in each 160 km by 160 km fitting window. The five waves are determined one by one. They are sorted with decreasing amplitudes. That is, the first wave at each grid point is the largest, and the fifth is the smallest. In most cases, only three waves are sufficient to account for >95% variance. We have tested fitting six or seven waves. But the sixth and seventh waves are usually lower than model noise level (see Figure 4c for model errors).

2. Is the plane wave fitting applicable when the wavenumber of the internal tide changes rapidly over variable bottom topography?

Good point. My plane wave analysis assumes simple plane waves in fitting windows of 160 km by 160 km (this study). This assumption is perfect in the open ocean. However, it is not a perfect representor in source regions due to complex topographic feature. This may lead to underestimation of internal tide amplitudes. This problem can be partly solved using smaller and smaller fitting windows (now limited by the wide ground tracks of nadir-looking altimeters). For example, 40 km by 40 km windows are used for nonrepeat altimetry missions (Zhao 2022 JPO). However, smaller fitting windows also lead to larger model errors.

3. Bandpass filtering in the wavenumber space: The barotropic tides may have horizontal scales comparable to those of the mode-1 internal tides. Does this matter? Also, what happens when the mode-2 and higher mode internal tides have large amplitudes near the generation sites?

Mode-1 $N_2$ internal tides have wavelengths typical of 150 km. In contrast, $N_2$ barotropic tides have wavelengths >3000 km. Mode-2 $N_2$ internal tides have wavelengths <80 km. They can be easily separated by any 2D bandpass filter.

4. What happens if the larger-amplitude M2 and S2 internal tides are Doppler-shifted to the N2 tidal frequency by time-varying background fields?

This is very interesting and challenging question. I spent a lot of time to prepare my answer. The answer is that our mapping of $N_2$ internal tides is not affected by the presence of $M_2$ internal tides. It is because $N_2$ and $M_2$ have different periods (12.6583 and 12.4206 hours), the 27-year-long SSH time series are sufficiently to separate them in the frequency domain.

My answer is supported by my work below:
First, I map $N_2$ internal tides using two data sets. One is the original 27-year-long data. The other is the same data but $M_2$ internal tides are predicted and subtracted using my $M_2$ internal tide model. I find that the resulting two $N_2$ internal tide models are almost the same.

Second, I map 13 sets of background internal tides between $M_2$ and $N_2$. Their tidal periods are linearly interpolated (with 1-min intervals) between $M_2$ and $N_2$. The 13 sets of internal tides are uncorrelated and just noise---they have no relation with the $N_2$ or $M_2$ internal tides. Since $M_2$ and $N_2$ cannot affect the 13 tidal periods between them, $M_2$ and $N_2$ internal tides must be independent with each other.

5. Do the results depend on the choice of several arbitrary parameters (such as the size of the fitting window)?

I want to stress that the parameters are chosen empirically, not arbitrary. In this study, I fit five waves (as explained above) in 160 km by 160 km window (as explained above) using a bandwidth of [0.8 1.25] times local wavenumber (explained in my reply to Reviewer #2). My empirical model is surely affected by these selections, but not much. And the degree of their influence varies from one region to another. Among these parameters, the large fitting window is an issue. I think I can continue to improve my model by fine adjusting these parameters (and thus seek maxima variance reduction).

6. Although the results presented in the manuscript seem reasonable overall, it is unclear what are the new findings in terms of the internal tide dynamics. Now the manuscript reads more like a progress report.

My initial plan is to publish a short note to simply present my new $N_2$ model. Now I change it to a full-length paper. The revised manuscript now has 13 figures (previously 4 figures). Now it presents new scientific findings in Figures 10-13. For example, Figure 12 shows that $N_2$ internal

tides are around the New Caledonia, one SWOT calibration/validation site proposed by French scientists. To my knowledge, this feature has been reported in the literature.

---

## Author Response (AR1)

Dear Dr. Katsumata,

Thank you very much for your encouragement and patience.

Following suggestions from three reviewers, I have completed a major revision of my work on mode-1 $N_2$ internal tides from satellite altimetry. I did a number of extra analyses to answer some interesting and challenging questions. Now I feel confident with my work and manuscript. I will continue to improve it with more comments from our reviewers.

However, because I did a major revision (almost a rewritten), it makes a changes-tracking file meaningless. Thus, I do not have a changes-tracking file. For next round, hopefully, it will be a minor revision, I will provide a changes-tracking file.

Thank you very much for your time and help!

Best regards,
Zhongxiang

---

## Author Response (AR2)

**Referee #2**

This manuscript describes the results of using a two-stage plane-wave-fitting technique to estimate the mode-1 baroclinic N2 tide from satellite altimeter data. The N2 tide is only about 20% as large as the M2 tide, so it represents a relatively small signal to be estimated, but, nonetheless, potentially significant in the interpretation of SWOT data in some regions.

The paper is a well-organized description of the author's methodology and results. Because the frequency of the N2 tide is close to that of M2, one largely expects the N2 fields to be very similar to those of M2, which are born out by the results. One novel aspect of this work is the author's technique for estimating the error in the harmonic constants by performing an analysis at nearby non-tidal frequencies where the expected results should consist entirely of noise. This technique has been used previously in other tidal studies, e.g., Ray and Susanto, 2016, and their work should be cited (search for "false" tides in that work). I think the author's use of the terminology "background tides" is confusing and he should find a better way to explain his error estimation procedure.

Thank you very much for your time and help!

In the revised manuscript, we cite two papers that employ similar techniques to estimate tidal errors: Ray and Susanto 2016; Zaron et al., 2023. Section 2.5 is heavily revised to better present our estimation procedure and resulting model errors.

Except for a few minor comments, below, I have no objection to publishing this manuscript in Ocean Science. I do think the results will be of limited interest in their current form, though, and with a few minor additions the interest and impact of this paper could be increased.

- The author computes an error estimate, but the estimate does not seem to be used in any way. The areas delimiting mesoscale contamination and western boundary currents seem to be taken from his previous work. Shouldn't the masking or delimiting areas of significance be based on the present error estimate?

In the revised Figures 3, 4 and 6, we delete the black contours taken from our previous papers.

Figure 6 is revised. It shows that the performance of the $N_2$ model can be related to the difference between the $N_2$ variance and 1.5 times the error variance. The factor of 1.5 is chosen, because it gives the best spatial correction between the variance difference (Figure 6c) and the $N_2$ variance reduction (Figure 6a). Figure 6d gives the mask indicating the region where the $N_2$ variance is greater than 1.5 times the error variance. The $N_2$ model performs better in the masked region.

- For comparisons with other estimates of the N2 tide which are likely to be produced by other groups, it would be useful if some quantitative summary statistics were presented, such as area-average potential or total energy of N2, and global average explained variance.

We have computed the globally integrated $N_2$ and $M_2$ energies following the method described in Zhao et al (2016 JPO). They are about 1.8 and 30.9 PJ, respectively. Their ratio is about 5.8%, larger than the theoretical value of 4%, because $N_2$ contains relatively more error variance (25%) than $M_2$ does (1%).

- The author's assertions that the tidal predictions will be useful for correcting SWOT data would have more impact if they were quantitative. For example, what is the expected explained variance averaged along the SWOT 1-day repeat tracks?

Along the SWOT daily repeat swaths, we have computed the variance reductions caused by the $N_2$ and $M_2$ internal tide models (ref. to Figures 6a and 6b). We have also computed the variance reductions in the masked region ($N_2$ variance > 1.5 error variance). The results show that both models perform better in the masked region. Please see our detailed description in Section 2.6.

- Personally, I thought much of Section 3 could be deleted or considerably shortened since the results are very much as would be expected based on our knowledge of M2.

We agree with the referee that "the results are very much as would be expected based on our knowledge of $M_2$." However, we can only expect the general features of $N_2$ internal tides from pre-known $M_2$ internal tides, due to our limited knowledge. More details should be provided by observations or numerical simulations. Figure 7a shows that the correlation coefficient between the $N_2$ and $M_2$ amplitudes is only 0.69, which suggests that we cannot accurately predict one tidal constituent from the other.

Itemized comments:

l25: Agreed that the N2 tide could cause variation in internal tide driven mixing; however, it might be worthwhile to point out how small this is expected to be in comparison with M2 (e.g., 4% if the mixing is a quadratic function of amplitude or 0.1% if the mixing is cubic). These variations are likely fall smaller than the uncertainty in the mixing caused by the larger components.

One sentence is added here: "*Theoretically, $N_2$ may modulate $M_2$ internal tides by ±20% in amplitude, and by ±40% in energy. On average, $N_2$ will enhance the $M_2$-induced ocean mixing by 4%.*"

l28: "provide" --> "could provide"

Changed.

l33: "it is" --> "it will be"

Changed as suggested.

l79: Since it was described in detail in previous studies, it should not be referred to as a "new" method in the abstract, etc.

Throughout the manuscript, 'new' is removed/changed in several places.

l102: Is there a reason why the rigid lid boundary condition is used? Doesn't this lead to errors of a few percent? (Wunsch and Chelton)

Changed. Our WOA18 phase speeds are computed using free-surface and rigid-bottom boundary conditions. In fact, our mapping procedure is not sensitive to the different boundary conditions.

l139-l146: This is a nice test for crosstalk.

Thanks.

l149: "in the our"

Fixed.

l151: "that"

Fixed.

l155: Is it fair to say that the rms error of the N2 tide is estimated to be 50%, or 25% error variance?

It is true. One sentence is added here: "O*n global average, the error variance is about 25% of the $N_2$ variance and 1% of the $M_2$ variance.*"

l159: Can you move the explanation of the term "background internal tides" to the start of the paragraph so the meaning is clearer?

This paragraph is rearranged to better present our estimation method.

l160: Why did you only use one of these estimates? Why not take mean amplitude across all the frequencies? Oh, I see, Figure 5 suggests that you only computed the error estimate at all frequencies within the regional box. Is that correct?

It is true. We run it in the central Pacific using 13 frequencies that evenly distribute between $N_2$ and $M_2$, and show that they give similar results. We then pick only one frequency (period is $N_2$ minus 3 minutes) for a global run (because it is time-consuming).

l217: "no shown"

Fixed.

l248: How was the calculation method "confirmed"?

Changed. Now it reads "*The same calculation method has also been derived by Geoffroy and Nycander (2022).*"

l253: Repetitive: "model results freely available"

This sentence is removed to avoid repetition.

Section 3: Overall, I thought that this section could be shortened considerably. I think it suffices to say that the N2 and M2 tides are similar, as expected. As for their differences, I am not sure much can be said, due to the small size and noise of the N2 maps. I do like the focus on regional maps, which are more intelligible than the global maps.

We agree with the referee that Section 3 is too long for the referee and readers who are familiar with this field (tide experts). However, our readers may have different research backgrounds. We want to show the detailed

regional and global maps to readers who are not very familiar with this topic and hope to see details to better understand our conclusions (intermediate researchers or graduate students or from related research fields).

Fig 6: You previously computed an error estimate in Figure 4. Would it be useful to mask off the explained variance estimate so that it excludes locations where the standard error is larger than, say, 50%, of the predicted amplitude? As-is, it appears that you have a lot of regions where the explained variance is either negative or unexpectedly large (e.g., in the WBC regions). It would be interesting to see how correlated are the explained variance and the predicted variance.

Figure 6 is revised. Figure 6c shows the difference between the $N_2$ variance and the error variance (times 1.5). Here a factor of 1.5 is chosen, because it gives the best spatial correlation between panels (a) and (c). Figure 6 suggests that the $N_2$ model performs well in the region where the $N_2$ variance is greater than 1.5 times the error variance. The masked region is shown in Figure 6d.

**Referee #3**

The revised manuscript now provides detailed descriptions of the method to extract mode-1 N2 internal tides and the characteristic features of the obtained internal tides. I appreciate the author's efforts to thoroughly revise the manuscript, which makes it easier for readers to see that the method is overall plausible. Although I still do not find any novelty in terms of the internal wave dynamics, the presented results seem reasonable and will make a solid contribution to the processing of the satellite altimetry data. Hence, I recommend publication of this manuscript after minor revisions.

Thank you very much for your time and help! Observing internal tides is the same important as exploring dynamics. Observations may indirectly improve our understanding of internal tide dynamics.

Minor Comments:

Response to Comment 3 of Reviewer #3: The author responded that the mode-2 N2 internal tides can be easily separated from the mode-1 N2 internal tides because they have significantly different wavelength. This is true, but I question why the author ignores the existence of mode-2 and higher modes even near the generation regions of internal tides, where higher modes may have comparatively large amplitude with mode-1. Note that the ultimate goal of this study is to remove all signals of internal tides from the satellite altimeter data for the study of mesoscale and sub-mesoscale processes.

We have tried mapping mode-2 $N_2$ internal tides around the Hawaiian Ridge (185–205ºE, 18–28ºN). However, the resulting model is noisy, as expected. In this region, the mean amplitude of mode-1 $N_2$ internal tides is about 2.5 mm. The mean mode-2 $N_2$ amplitude would be 1 mm, using a ratio of ~2.5 from mode-1 and -2 $M_2$ internal tides. The 1-mm mode-2 $N_2$ signal cannot overcome the ~0.7-mm error. This discussion is added to the summary section.

Response to Comment 4 of Reviewer #3: If M2 internal tides are randomly and weakly Doppler-shifted by time-varying background fields, the resulting internal wave frequency will spread centered around the M2 tidal frequency. Therefore, Fig. 5, which does not show such spreading in the frequency space, may be considered evidence that the Doppler-shifting effects are not very strong (at least in this particular region). However, I do not understand why the results described in Lines 139–146 can be regarded as evidence to show that the Doppler-shift is weak. If M2 internal tides are Doppler-shifted by time-varying background fields, they no longer have M2 tidal frequency even in a fixed frame of reference.

Our analysis described in Lines 139–146 is to show that "the $N_2$ and $M_2$ internal tides do not crosstalk in our mapping method." It is not to examine the Doppler effect of background currents on internal tides.

Section 2.3: Why does the author need to perform the plane wave analysis twice in the data processing (steps 1 and 3)? Does the obtained tidal field change significantly if one skips step 1? If so, why?

The first-round plane wave analysis is to map internal tides from along-track altimetry data to horizontal fields at a regular longitude-latitude grid. It cannot be skipped. The second-round plane wave analysis is to decompose internal tides into five waves of arbitrary directions at each grid point. It cannot be skipped either.

Lines 295–296: Barotropic-to-baroclinic energy conversion is not examined in this manuscript.

This sentence is repetitive and inaccurate, thus removed in revision.

---

## Author Response (AR3)

Thank you for revising to reflect comments from reviewers #2 and #3. I have a few questions regarding the revision. Could you please consider these points before finalizing this manuscript? Line numbers are those of the Track Change file (egusphere-2022-1029-ATC2.pdf).

Thank you for your time and encouragement.

L.6. mm-scale → millimeter-scale

Changed in three places.

L.31-32. I can see why $N_2$ can modify $M_2$ by 20 % in amplitude, but why 4 % of mixing? If mixing is a quadratic function of amplitude, I would argue it is 40 % larger (1.2 * 1.2 = 1.4). Or is it 4 % (0.2 * 0.2 = 0.04)?

It is 4% (0.2*0.2 = 0.04). Now the two sentences read: "*Theoretically, $N_2$ may modulate $M_2$ internal tides by ±20% in amplitude, and by ±40% in energy (i.e., $(1±0.2)^2$). On average, $N_2$ will enhance the $M_2$-induced ocean mixing by 4% (i.e., $0.2^2$).*"

Section 2.6, σ is usually used to denote standard deviation (square root of variance) not the variance. It might be less confusing to follow this tradition.

Thanks for pointing it out. We examine variance (not its square root) in this study; therefore, we replace σ with $σ^2$ throughout this manuscript to avoid confusion.

Considering that (1) σ was used in our reply to Ref#2, and (2) the best empirical factor (*m*) is 2 (not 1.5), we replot Figure 6 to better present our analysis process. Figure 6 now has six panels. The two new panels are for the model variance (c) and the error variance (d). Figure caption and relevant text are edited accordingly.

L.211 and L.214. The integrated variance reduction is difficult to interpret as there is no obvious quantity to compare these numbers with. Is 500 small, is $1.9×10^4$ large? It also depends on the integration path lengths (i.e. mask area in Fig.6(d)). The dimension appears wrong as it should have a dimension of $(length)^3$ after integration along the tracks. I am aware of the comment from Ref#2 relevant to this paragraph. I suppose a relative reduction rate (e.g. reduction of X%) would be sufficient.

Good point. In our previous along-track integration, we omitted the across-track width, and thus the results were in $mm^2$. Following this suggestion, we now calculate the along-track mean variance reductions (not integrals suggested by Ref#2).

We feel that a relative reduction rate (%) does not help, because internal tides are very weak signals in the satellite altimetry data. $N_2$ internal tides account for about 0.05% of the total SSH variance, which explains why it is challenging to extract $N_2$ internal tides. But keep in mind that internal tides and weak submesoscale motions have comparable variances and similar spatial scales, which partly motivates this study.

Caption to Fig.9, It is arguable whether the Antarctic Circumpolar Current is a "boundary" current or not.

Changed. Now it reads: "*Green contours indicate regions of strong mesoscale motions, where the $N_2$ internal tides are overwhelmed by errors (see Figure 6e).*"